# CRoSS: Diffusion Model Makes Controllable, Robust and Secure Image Steganography

**Jiwen Yu**[1]  **Xuanyu Zhang**[1]  **Youmin Xu**[1,2]  **Jian Zhang**[1†]

[1] Peking University Shenzhen Graduate School  [2] Peng Cheng Laboratory

## Abstract

Current image steganography techniques are mainly focused on cover-based methods, which commonly have the risk of leaking secret images and poor robustness against degraded container images. Inspired by recent developments in diffusion models, we discovered that two properties of diffusion models, the ability to achieve translation between two images without training, and robustness to noisy data, can be used to improve security and natural robustness in image steganography tasks. For the choice of diffusion model, we selected Stable Diffusion, a type of conditional diffusion model, and fully utilized the latest tools from open-source communities, such as LoRAs and ControlNets, to improve the controllability and diversity of container images. In summary, we propose a novel image steganography framework, named **C**ontrollable, **Ro**bust and **S**ecure Image **S**teganography (CRoSS), which has significant advantages in controllability, robustness, and security compared to cover-based image steganography methods. These benefits are obtained without additional training. To our knowledge, this is the first work to introduce diffusion models to the field of image steganography. In the experimental section, we conducted detailed experiments to demonstrate the advantages of our proposed CRoSS framework in controllability, robustness, and security. Code is available at `https://github.com/vvictoryuki/CRoSS`.

## 1 Introduction

With the explosive development of digital communication and AIGC (AI-generated content), the privacy, security, and protection of data have aroused significant concerns. As a widely studied technique, steganography [10] aims to hide messages like audio, image, and text into the container image in an undetected manner. In its reveal process, it is only possible for the receivers with pre-defined revealing operations to reconstruct secret information from the container image. It has a wide range of applications, such as copyright protection [4], digital watermarking [15], e-commerce [11], anti-visual detection [34], spoken language understanding [12, 13] and cloud computing [76].

For image steganography, traditional methods tend to transform the secret messages in the spatial or adaptive domains [27], such as fewer significant bits [9] or indistinguishable parts. With the development of deep neural networks, researchers begin to use auto-encoder networks [5, 6] or invertible neural networks (INN) [35, 26]to hide data, namely deep steganography.

The essential targets of image steganography are security, reconstruction quality, and robustness [9, 45, 77]. Since most previous methods use cover images to hide secret images, they tend to explicitly retain some secret information as artifacts or local details in the container image, which poses a risk of information leakage and reduces the **security** of transmission. Meanwhile, although previous works can maintain well reconstruction fidelity of the revealed images, they tend to train models in a noise-free simulation environment and can not withstand noise, compression artifacts, and non-linear transformations in practice, which severely hampers their practical values and **robustness** [30, 44, 25].

---

[†]Corresponding author. This work was supported by National Natural Science Foundation of China under Grant 62372016.

To address security and robustness concerns, researchers have shifted their focus toward coverless steganography. This approach aims to create a container image that bears no relation to the secret information, thereby enhancing its security. Current coverless steganography methods frequently employ frameworks such as CycleGAN [78] and encoder-decoder models [76], leveraging the principle of cycle consistency. However, the **controllability** of the container images generated by existing coverless methods remains limited. Their container images are only randomly sampled from the generative model and cannot be determined by the user. Moreover, existing approaches [47] tend to only involve hiding bits into container images, ignoring the more complex hiding of secret images. Overall, current methods, whether cover-based or coverless, have not been able to achieve good unity in terms of security, controllability, and robustness. Thus, our focus is to propose a new framework that can simultaneously improve existing methods in these three aspects.

Recently, research on diffusion-based generative models [22, 55, 56] has been very popular, with various unique properties such as the ability to perform many tasks in a zero-shot manner [36, 28, 64, 63, 72, 37, 20], strong control over the generation process [16, 49, 74, 40, 19, 50], natural robustness to noise in images [64, 28, 14, 65], and the ability to achieve image-to-image translation [75, 8, 20, 39, 57, 14, 29, 37]. We were pleasantly surprised to find that these properties perfectly match the goals we mentioned above for image steganography: (1) **Security**: By utilizing the DDIM Inversion technique [54] for diffusion-based image translation, we ensure the invertibility of the translation process. This invertible translation process enables a coverless steganography framework, ensuring the security of the hidden image. (2) **Controllability**: The powerful control capabilities of conditional diffusion models make the container image highly controllable, and its visual quality is guaranteed by the generative prior of the diffusion model; (3) **Robustness**: Diffusion models are essentially Gaussian denoisers and have natural robustness to noise and perturbations. Even if the container image is degraded during transmission, we can still reveal the main content of the secret image.

We believe that the fusion of diffusion models and image steganography is not simply a matter of mechanically combining them, but rather an elegant and instructive integration that takes into account the real concerns of image steganography. Based on these ideas, we propose the **C**ontrollable, **Ro**bust and **S**ecure Image **S**teganography (**CRoSS**) framework, a new image steganography framework that aims to simultaneously achieve gains in security, controllability, and robustness.

Our contributions can be summarized as follows:

- We identify the limitations of existing image steganography methods and propose a unified goal of achieving security, controllability, and robustness. We also demonstrate that the diffusion model can seamlessly integrate with image steganography to achieve these goals using diffusion-based invertible image translation technique without requiring any additional training.

- We propose a new image steganography framework: Controllable, Robust and Secure Image Steganography (CRoSS). To the best of our knowledge, this is the first attempt to apply the diffusion model to the field of image steganography and gain better performance.

- We leveraged the progress of the rapidly growing Stable Diffusion community to propose variants of CRoSS using prompts, LoRAs, and ControlNets, enhancing its controllability and diversity.

- We conducted comprehensive experiments focusing on the three targets of security, controllability, and robustness, demonstrating the advantages of CRoSS compared to existing methods.

## 2 Related Work

### 2.1 Steganography Methods

**Cover-based Image Steganography.** Unlike cryptography, steganography aims to hide secret data in a host to produce an information container. For image steganography, a cover image is required to hide the secret image in it [5]. Traditionally, spatial-based [24, 41, 43, 46] methods utilize the Least Significant Bits (LSB), pixel value differencing (PVD) [43], histogram shifting [60], multiple bit-planes [41] and palettes [24, 42] to hide images, which may arise statistical suspicion and are vulnerable to steganalysis methods. Adaptive methods [45, 31] decompose the steganography into embedding distortion minimization and data coding, which is indistinguishable by appearance but limited in capacity. Various transform-based schemes [10, 27] including JSteg [46] and DCT steganography [21] also fail to offer high payload capacity. Recently, various deep learning-based

schemes have been proposed to solve image steganography. Baluja [5] proposed the first deep-learning method to hide a full-size image into another image. Generative adversarial networks (GANs) [53] are introduced to synthesize container images. Probability map methods focus on generating various cost functions satisfying minimal-distortion embedding [45, 59]. [69] proposes a generator based on U-Net. [58] presents an adversarial scheme under distortion minimization. Three-player game methods like SteganoGAN [73] and HiDDeN [77] learn information embedding and recovery by auto-encoder architecture to adversarially resist steganalysis. Recent attempts [66] to introduce invertible neural networks (INN) into low-level inverse problems like denoising, rescaling, and colorization show impressive potential over auto-encoder, GAN [3], and other learning-based architectures. Recently, [35, 26] proposed designing the steganography model as an invertible neural network (INN) [17, 18] to perform image hiding and recovering with a single INN model.

**Coverless Steganography.** Coverless steganography is an emerging technique in the field of information hiding, which aims to embed secret information within a medium without modifying the cover object [47]. Unlike traditional steganography methods that require a cover medium (e.g., an image or audio file) to be altered for hiding information, coverless steganography seeks to achieve secure communication without introducing any changes to the cover object [33]. This makes it more challenging for adversaries to detect the presence of hidden data, as there are no observable changes in the medium's properties [38]. To the best of our knowledge, existing coverless steganography methods [34] still focus on hiding bits into container images, and few explorations involve hiding images without resorting to cover images.

## 2.2 Diffusion Models

Diffusion models [22, 55, 56] are a type of generative model that is trained to learn the target image distribution from a noise distribution. Recently, due to their powerful generative capabilities, diffusion models have been widely used in various image applications, including image generation [16, 48, 51, 49], restoration [52, 28, 64], translation [14, 29, 37, 75], and more. Large-scale diffusion model communities have also emerged on the Internet, with the aim of promoting the development of AIGC(AI-generated content)-related fields by applying the latest advanced techniques.

In these communities, the Stable Diffusion [49] community is currently one of the most popular and thriving ones, with a large number of open-source tools available for free, including model checkpoints finetuned on various specialized datasets. Additionally, various LoRAs [23] and ControlNets [74] are available in these communities for efficient control over the results generated by Stable Diffusion. LoRAs achieve control by efficiently modifying some network parameters in a low-rank way, while ControlNets introduce an additional network to modify the intermediate features of Stable Diffusion for control. These mentioned recent developments have enhanced our CRoSS framework.

# 3 Method

## 3.1 Definition of Image Steganography

Before introducing our specific method, we first define the image steganography task as consisting of three images and two processes (as shown in Fig. 1): the three images refer to the secret image $\mathbf{x}_{sec}$, container image $\mathbf{x}_{cont}$, and revealed image $\mathbf{x}_{rev}$, while the two processes are the hide process and reveal process. The secret image $\mathbf{x}_{sec}$ is the image we want to hide and is hidden in the container image $\mathbf{x}_{cont}$ through the hide process. After transmission over the Internet, the container image $\mathbf{x}_{cont}$ may become degraded, resulting in a degraded container image $\mathbf{x}'_{cont}$, from which we extract the revealed image $\mathbf{x}_{rev}$ through the reveal process.

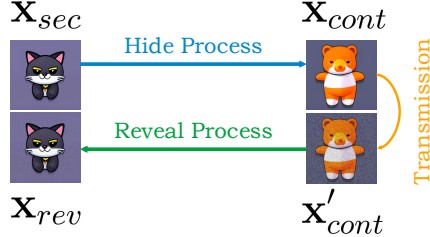

Figure 1: Illustration used to show the definition of image steganography.

Our goal is to make our proposed framework have the following properties: (1) **Security**: even if the container image $\mathbf{x}_{cont}$ is intercepted by other receivers, the hidden secret image $\mathbf{x}_{sec}$ cannot be leaked. (2) **Controllability**: the content in the container image $\mathbf{x}_{cont}$ can be controlled by the user, and its visual quality is high. (3) **Robustness**: the reveal process can still generate semantically consistent results ($\mathbf{x}_{rev} \approx \mathbf{x}_{sec}$) even if there is deviation in the $\mathbf{x}'_{cont}$ compared to the $\mathbf{x}_{cont}$ ($\mathbf{x}'_{cont} = d(\mathbf{x}_{cont})$, $d(\cdot)$ denotes the degradation process). According to the above definition,

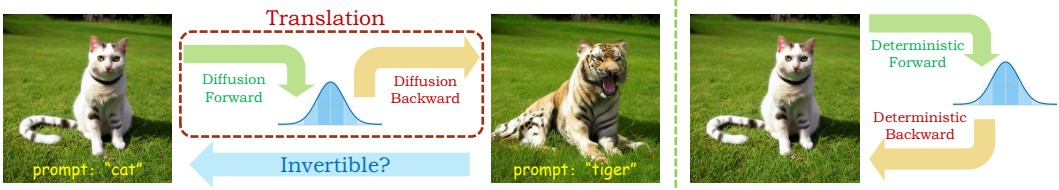

| (a) Image Translation using Different Conditions | (b) DDIM Inversion |

Figure 2: In part (a), Conditional diffusion models can be used with different conditions to perform image translation. In this example, we use two different prompts ("cat" and "tiger") to translate a cat image into a tiger image. However, a critical challenge for coverless image steganography is whether we can reveal the original image from the translated image. The answer is yes, and we can use DDIM Inversion (shown in part (b)) to achieve dual-direction translation between the image distribution and noise distribution, allowing for invertible image translation.

we can consider the hide process as a translation between the secret image $\mathbf{x}_{sec}$ and the container image $\mathbf{x}_{cont}$, and the reveal process as the inverse process of the hide process. In Sec. 3.2, we will introduce how to use diffusion models to implement these ideas, and in Sec. 3.3, we will provide a detailed description of our proposed framework CRoSS for coverless image steganography.

## 3.2 Invertible Image Translation using Diffusion Model

**Diffusion Model Defined by DDIM.** A complete diffusion model process consists of two stages: the forward phase adds noise to a clean image, while the backward sampling phase denoises it step by step. In DDIM [54], the formula for the forward process is given by:

$$\mathbf{x}_t = \sqrt{\alpha_t}\mathbf{x}_{t-1} + \sqrt{1-\alpha_t}\boldsymbol{\epsilon}, \quad \boldsymbol{\epsilon} \sim \mathcal{N}(\mathbf{0}, \mathbf{I}), \tag{1}$$

where $\mathbf{x}_t$ denotes the noisy image in the $t$-th step, $\boldsymbol{\epsilon}$ denotes the randomly sampled Gaussian noise, $\alpha_t$ is a predefined parameter and the range of time step $t$ is $[1, T]$. The formula of DDIM for the backward sampling process is given by:

$$\mathbf{x}_s = \sqrt{\bar{\alpha}_s}\mathbf{f}_{\boldsymbol{\theta}}(\mathbf{x}_t, t) + \sqrt{1-\bar{\alpha}_s-\sigma_s^2}\boldsymbol{\epsilon}_{\boldsymbol{\theta}}(\mathbf{x}_t, t) + \sigma_s\boldsymbol{\epsilon}, \quad \mathbf{f}_{\boldsymbol{\theta}}(\mathbf{x}_t, t) = \frac{\mathbf{x}_t - \sqrt{1-\bar{\alpha}_t}\boldsymbol{\epsilon}_{\boldsymbol{\theta}}(\mathbf{x}_t, t)}{\sqrt{\bar{\alpha}_t}}, \tag{2}$$

where $\boldsymbol{\epsilon} \sim \mathcal{N}(\mathbf{0}, \mathbf{I})$ is a randomly sampled Gaussian noise with $\sigma_s^2$ as the noise variance, $\mathbf{f}_{\boldsymbol{\theta}}(\cdot, t)$ is a denoising function based on the pre-trained noise estimator $\boldsymbol{\epsilon}_{\boldsymbol{\theta}}(\cdot, t)$, and $\bar{\alpha}_t = \prod_{i=1}^{t} \alpha_i$. DDIM does not require the two steps in its sampling formula to be adjacent (i.e., $t = s + 1$). Therefore, $s$ and $t$ can be any two steps that satisfy $s < t$. This makes DDIM a popular algorithm for accelerating sampling. Furthermore, if $\sigma_s$ in Eq.2 is set to 0, the DDIM sampling process becomes deterministic. In this case, the sampling result is solely determined by the initial value $\mathbf{x}_T$, which can be considered as a latent code. The sampling process can also be equivalently described as solving an Ordinary Differential Equation (ODE) using an ODE solver [54]. In our work, we choose deterministic DDIM to implement the diffusion model and use the following formula:

$$\mathbf{x}_0 = \text{ODESolve}(\mathbf{x}_T; \boldsymbol{\epsilon}_{\boldsymbol{\theta}}, T, 0) \tag{3}$$

to represent the process of sampling from $\mathbf{x}_T$ to $\mathbf{x}_0$ using a pretrained noise estimator $\boldsymbol{\epsilon}_{\boldsymbol{\theta}}$.

**Image Translation using Diffusion Model.** A large number of image translation methods [75, 8, 20, 39, 57, 14, 29, 37] based on diffusion models have been proposed. In our method, we will adopt a simple approach. First, we assume that the diffusion models used in our work are all conditional diffusion models that support condition $\mathbf{c}$ as input to control the generated results. Taking the example shown in Fig. 2 (a), suppose we want to transform an image of a cat into an image of a tiger. We add noise to the cat image using the forward process (Eq. 1) to obtain the intermediate noise, and then control the backward sampling process (Eq. 2) from noise by inputting a condition (prompt="tiger"), resulting in a new tiger image. In general, if the sampling condition is set to $\mathbf{c}$, our conditional sampling process can be expressed based on Eq. 3 as follows:

$$\mathbf{x}_0 = \text{ODESolve}(\mathbf{x}_T; \boldsymbol{\epsilon}_{\boldsymbol{\theta}}, \mathbf{c}, T, 0). \tag{4}$$

For image translation, there are two properties that need to be considered: the structural consistency of the two images before and after the translation, and whether the translation process is invertible.

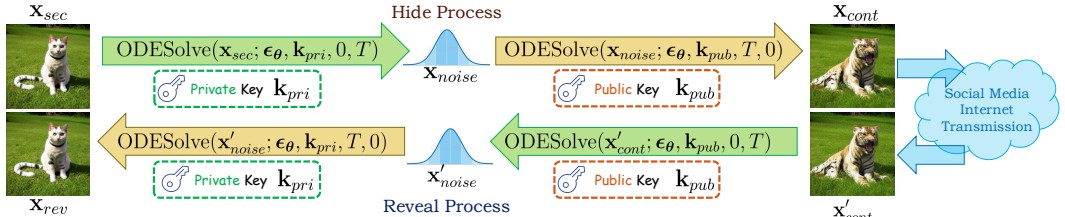

Figure 3: Our coverless image steganography framework CRoSS. The diffusion model we choose is a conditional diffusion model, which supports conditional inputs to control the generation results. We choose the deterministic DDIM as the sampling strategy and use the two different conditions ($\mathbf{k}_{pri}$ and $\mathbf{k}_{pub}$) given to the model as the private key and the public key.

---

**Algorithm 1** The Hide Process of CRoSS.

---

**Input:** The secret image $\mathbf{x}_{sec}$ which will be hidden, a pre-trained conditional diffusion model with noise estimator $\epsilon_{\boldsymbol{\theta}}$, the number $T$ of time steps for sampling and two different conditions $\mathbf{k}_{pri}$ and $\mathbf{k}_{pub}$ which serve as the private and public keys.
**Output:** The container image $\mathbf{x}_{cont}$ used to hide the secret image $\mathbf{x}_{sec}$.
$\mathbf{x}_{noise} = \text{ODESolve}(\mathbf{x}_{sec}; \epsilon_{\boldsymbol{\theta}}, \mathbf{k}_{pri}, 0, T)$
$\mathbf{x}_{cont} = \text{ODESolve}(\mathbf{x}_{noise}; \epsilon_{\boldsymbol{\theta}}, \mathbf{k}_{pub}, T, 0)$
**return** $\mathbf{x}_{cont}$

---

Structural consistency is crucial for most applications related to image translation, but for coverless image steganography, ensuring the invertibility of the translation process is the more important goal. To achieve invertible image translation, we utilize DDIM Inversion based on deterministic DDIM.

**DDIM Inversion Makes an Invertible Image Translation.** DDIM Inversion (shown in Fig. 2 (b)), as the name implies, refers to the process of using DDIM to achieve the conversion from an image to a latent noise and back to the original image. The idea is based on the approximation of forward and backward differentials in solving ordinary differential equations [54, 29]. Intuitively, in the case of deterministic DDIM, it allows $s$ and $t$ in Eq. 2 to be any two steps (i.e., allowing $s < t$ and $s > t$). When $s < t$, Eq. 2 performs the backward process, and when $s > t$, Eq. 2 performs the forward process. As the trajectories of forward and backward processes are similar, the input and output images are very close, and the intermediate noise $\mathbf{x}_T$ can be considered as the latent variable of the inversion. In our work, we use the following formulas:

$$\mathbf{x}_T = \text{ODESolve}(\mathbf{x}_0; \epsilon_{\boldsymbol{\theta}}, \mathbf{c}, 0, T), \quad \mathbf{x}'_0 = \text{ODESolve}(\mathbf{x}_T; \epsilon_{\boldsymbol{\theta}}, \mathbf{c}, T, 0), \tag{5}$$

to represent the DDIM Inversion process from the original image $\mathbf{x}_0$ to the latent code $\mathbf{x}_T$ and from the latent code $\mathbf{x}_T$ back to the original image $\mathbf{x}_0$ (the output image is denoted as $\mathbf{x}'_0$ and $\mathbf{x}'_0 \approx \mathbf{x}_0$). Based on DDIM Inversion, we have achieved the invertible relationship between images and latent noises. As long as we use deterministic DDIM to construct the image translation framework, the entire framework can achieve invertibility with two DDIM Inversion loops. It is the basis of our coverless image steganography framework, which will be described in detail in the next subsection.

### 3.3 The Coverless Image Steganography Framework CRoSS

Our basic framework CRoSS is based on a conditional diffusion model, whose noise estimator is represented by $\epsilon_{\boldsymbol{\theta}}$, and two different conditions that serve as inputs to the diffusion model. In our work, these two conditions can serve as the private key and public key (denoted as $\mathbf{k}_{pri}$ and $\mathbf{k}_{pub}$), as shown in Fig.3, with detailed workflow described in Algo.1 and Algo. 2. We will introduce the entire CRoSS framework in two parts: the hide process and the reveal process.

**The Process of Hide Stage.** In the hide stage, we attempt to perform translation between the secret image $\mathbf{x}_{sec}$ and the container image $\mathbf{x}_{cont}$ using the forward and backward processes of deterministic DDIM. In order to make the images before and after the translation different, we use the pre-trained conditional diffusion model with different conditions in the forward and backward processes respectively. These two different conditions also serve as private and public keys in the CRoSS framework. Specifically, the private key $\mathbf{k}_{pri}$ is used for the forward process, while the

**Algorithm 2** The Reveal Process of CRoSS.

---

**Input:** The container image $\mathbf{x}'_{cont}$ that has been transmitted over the Internet (may be degraded from $\mathbf{x}_{cont}$), the pre-trained conditional diffusion model with noise estimator $\epsilon_\theta$, the number $T$ of time steps for sampling, the private key $\mathbf{k}_{pri}$ and public key $\mathbf{k}_{pub}$.
**Output:** The revealed image $\mathbf{x}_{rev}$.
$\mathbf{x}'_{noise} = \text{ODESolve}(\mathbf{x}'_{cont}; \epsilon_\theta, \mathbf{k}_{pub}, 0, T)$
$\mathbf{x}_{rev} = \text{ODESolve}(\mathbf{x}'_{noise}; \epsilon_\theta, \mathbf{k}_{pri}, T, 0)$
**return** $\mathbf{x}_{rev}$

---

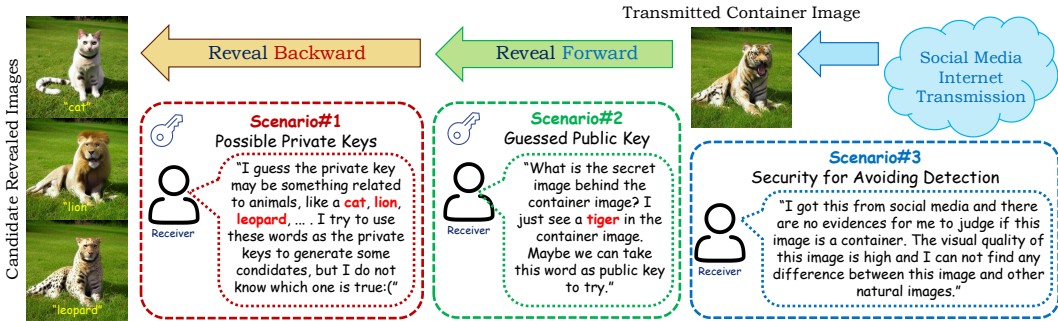

Figure 4: Further explanation of the CRoSS framework. We simulated the possible problems that a receiver may encounter in three different scenarios during the reveal process.

public key $\mathbf{k}_{pub}$ is used for the backward process. After getting the container image $\mathbf{x}_{cont}$, it will be transmitted over the Internet and publicly accessible to all potential receivers.

**The Roles of the Private and Public Keys in Our CRoSS Framework.** In CRoSS, we found that these given conditions can act as keys in practical use. The private key is used to describe the content in the secret image, while the public key is used to control the content in the container image. For the public key, it is associated with the content in the container image, so even if it is not manually transmitted over the network, the receiver can guess it based on the received container image (described in Scenario#2 of Fig. 4). For the private key, it determines whether the receiver can successfully reveal the original image, so it cannot be transmitted over public channels.

**The Process of Reveal Stage.** In the reveal stage, assuming that the container image has been transmitted over the Internet and may have been damaged as $\mathbf{x}'_{cont}$, the receiver needs to reveal it back to the secret image through the same forward and backward process using the same conditional diffusion model with corresponding keys. Throughout the entire coverless image steganography process, we do not train or fine-tune the diffusion models specifically for image steganography tasks but rely on the inherent invertible image translation guaranteed by the DDIM Inversion.

**The Security Guaranteed by CRoSS.** Some questions about security may be raised, such as: What if the private key is guessed by the receivers? Does the container image imply the possible hidden secret image? We clarify these questions from two aspects: (1) Since the revealed image is generated by the diffusion model, the visual quality of the revealed image is relatively high regardless of whether the input private key is correct or not. The receiver may guess the private key by exhaustive method, but it is impossible to judge which revealed image is the true secret image from a pile of candidate revealed images (described in Scenario#1 of Fig. 4). (2) Since the container image is also generated by the diffusion model, its visual quality is guaranteed by the generative prior of the diffusion model. Moreover, unlike cover-based methods that explicitly store clues in the container image, the container image in CRoSS does not contain any clues that can be detected or used to extract the secret image. Therefore, it is hard for the receiver to discover that the container image hides other images or to reveal the secret image using some detection method (described in Scenario#3 of Fig. 4).

**Various Variants for Public and Private Keys.** Our proposed CRoSS relies on pre-trained conditional diffusion models with different conditions $\mathbf{k}_{pub}, \mathbf{k}_{pri}$ and these conditions serve as keys in the CRoSS framework. In practical applications, we can distinguish different types of conditions of diffusion models in various ways. Here are some examples: (1) **Prompts**: using the same checkpoint

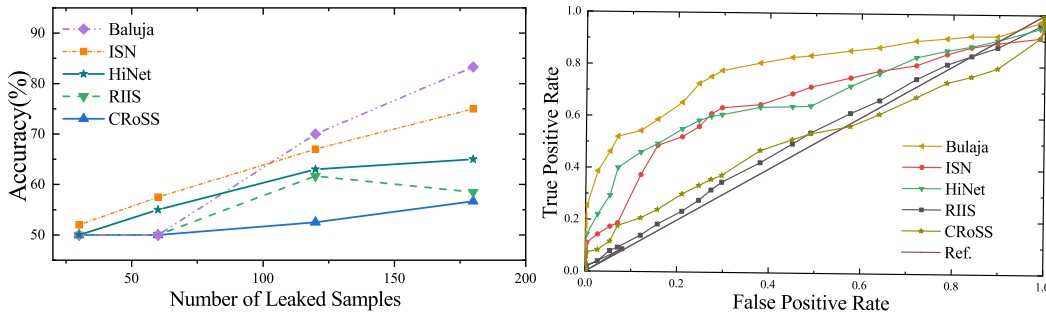

Figure 5: Deep steganalysis results by the latest SID [61]. As the number of leaked samples increases, methods whose detection accuracy curves grow more slowly and approach $50\%$ exhibit higher security. The right is the recall curve of different methods under the StegExpose [7] detector. The closer the area enclosed by the curve and the coordinate axis is to 0.5, the closer the method is to the ideal evasion of the detector.

of text-to-image diffusion models like Stable Diffusion [49] but different prompts as input conditions; (2) **LoRAs** [23]: using the same checkpoint initialization, but loading different LoRAs; (3) **ControlNets** [74]: loading the same checkpoint but using ControlNet with different conditions.

## 4 Experiment

### 4.1 Implementation Details

**Experimental Settings.** In our experiment, we chose Stable Diffusion [49] v1.5 as the conditional diffusion model, and we used the deterministic DDIM [54] sampling algorithm. Both the forward and backward processes consisted of 50 steps. To achieve invertible image translation, we set the guidance scale of Stable Diffusion to 1. For the given conditions, which serve as the private and public keys, we had three options: prompts, conditions for ControlNets [74] (depth maps, scribbles, segmentation maps), and LoRAs [23]. All experiments were conducted on a GeForce RTX 3090 GPU card, and our method did not require any additional training or fine-tuning for the diffusion model. The methods we compared include RIIS [68], HiNet [26], Baluja [6], and ISN [35].

**Data Preparation.** To perform a quantitative and qualitative analysis of our method, we collect a benchmark with a total of 260 images and generate corresponding prompt keys specifically tailored for the coverless image steganography, dubbed Stego260. We categorize the dataset into three classes, namely humans, animals, and general objects (such as architecture, plants, food, furniture, etc.). The images in the dataset are sourced from publicly available datasets [1, 2] and Google search engines. For generating prompt keys, we utilize BLIP [32] to generate private keys and employ ChatGPT or artificial adjustment to perform semantic modifications and produce public keys in batches. More details about the dataset can be found in the supplementary material.

### 4.2 Property Study#1: Security

In Fig. 5, the recent learning-based steganalysis method Size-Independent-Detector (SID) [61] is retrained with leaked samples from testing results of various methods on Stego260. The detection accuracy of CRoSS increases more gradually as the number of leaked samples rises, compared to other methods. The recall curves on the right also reveal the lower detection accuracy of our CRoSS, indicating superior anti-steganalysis performance.

Our security encompasses two aspects: imperceptibility in visual quality against human suspicion and resilience against steganalysis attacks. NIQE is a no-reference image quality assessment (IQA) model to measure the naturalness and visual

| Methods | NIQE↓ | |Detection Accuracy - 50|↓ | | |
|---|---|---|---|---|
| | | XuNet [67] | YedroudjNet [70] | KeNet [71] |
| Baluja [6] | 3.43±0.08 | 45.18±1.69 | 43.12±2.18 | 46.88±2.37 |
| ISN [35] | 2.87±0.02 | 5.14±0.44 | 3.01±0.29 | 8.62±1.19 |
| HiNet [26] | 2.94±0.02 | 5.29±0.44 | 3.12±0.36 | 8.33±1.22 |
| RIIS [68] | 3.13±0.05 | 0.73±0.13 | 0.24±0.08 | 4.88±1.15 |
| CRoSS (ours) | 3.04 | 1.32 | 0.18 | 2.11 |

Table 1: Security analysis. NIQE indicates the visual quality of container images, lower is better. The closer the detection rate of a method approximates $50\%$, the more secure the method is considered, as it suggests its output is indistinguishable from random chance. The best results are red and the second-best results are blue.

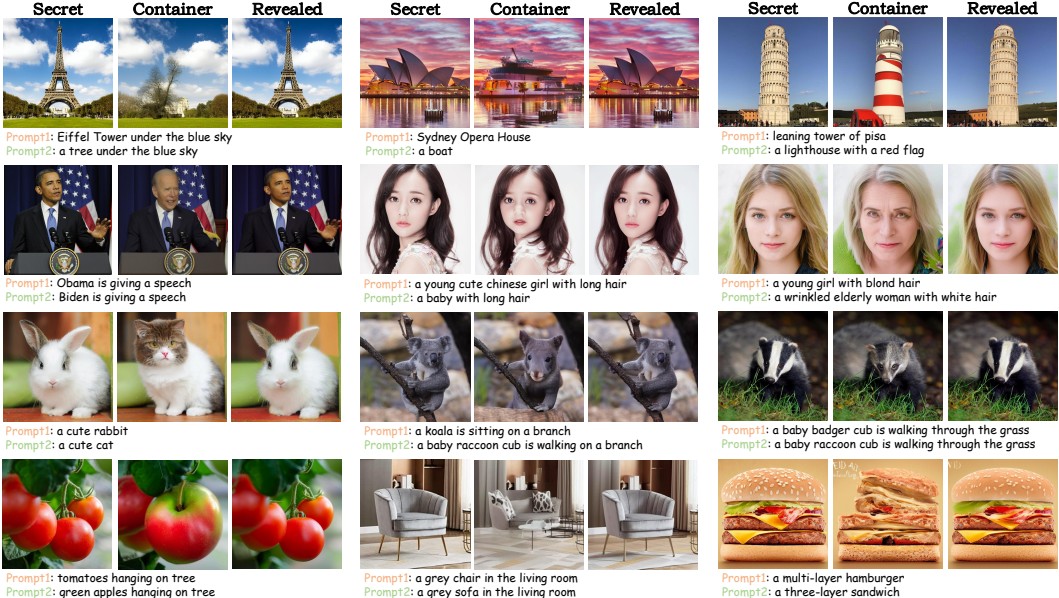

Figure 6: Visual results of the proposed CRoSS controlled by different prompts. The container images are realistic and the revealed images have well semantic consistency with the secret images.

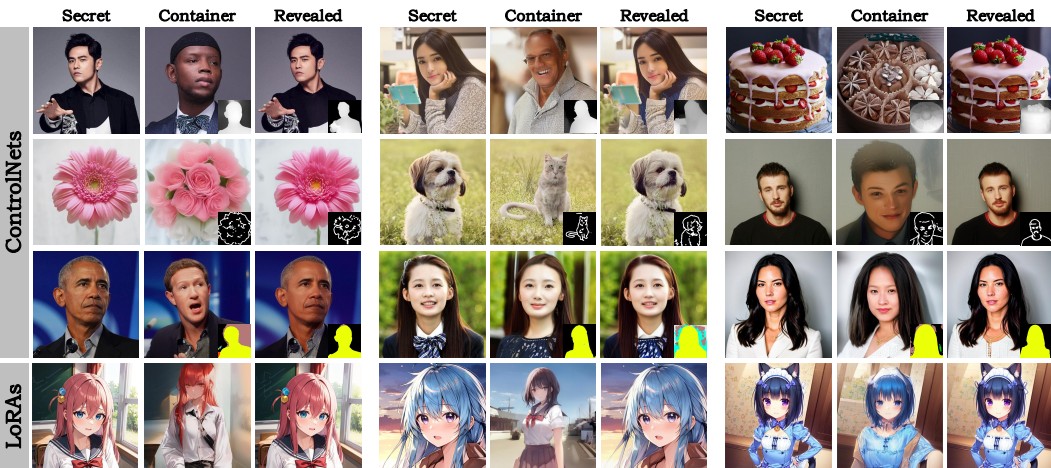

Figure 7: Visual results of our CRoSS controlled by different ControlNets and LoRAs. Depth maps, scribbles, and segmentation maps are presented in the lower right corner of the images.

security without any reference image or human feedback. In Tab. 1, the lower the NIQE score, the less likely it is for the human eye to identify the image as a potentially generated container for hiding secret information. Our NIQE is close to those of other methods, as well as the original input image (2.85), making it difficult to discern with human suspicion. Anti-analysis security is evaluated by three steganalysis models XuNet[67], YedroudjNet[70], and KeNet[71], for which lower detection accuracy denotes higher security. Our CRoSS demonstrates the highest or near-highest resistance against various steganalysis methods.

## 4.3 Property Study#2: Controllability

To verify the controllability and flexibility of the proposed CRoSS, various types of private and public keys such as prompts, ControlNets, and LoRAs † are incorporated in our framework. As illustrated in Fig. 6, our framework is capable of effectively hiding the secret images in the container images based on the user-provided "Prompt2" without noticeable artifacts or unrealistic image details. The container image allows for the seamless modification of a person's identity information, facial attributes, as

---

†The last row of Fig. 7 are generated via LoRAs downloaded from https://civitai.com/.

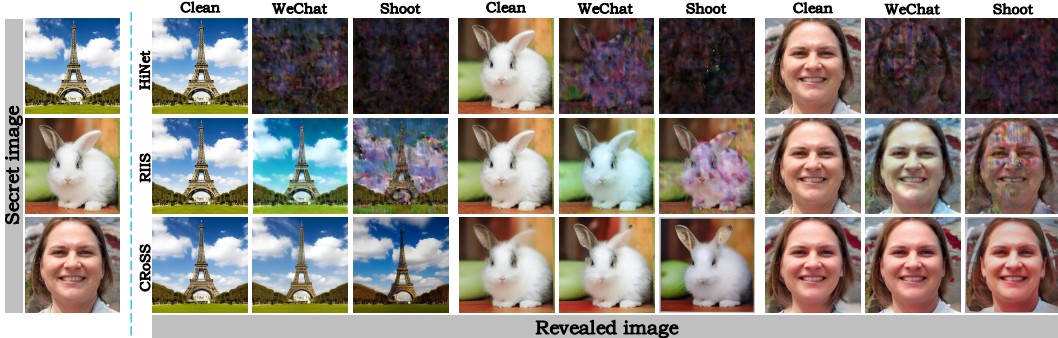

Figure 8: Visual comparisons of our CRoSS and other methods [68, 26] under two real-world degradations, namely "WeChat" and "Shoot". Obviously, our method can reconstruct the content of secret images, while other methods exhibit significant color distortion or have completely failed.

| Methods | clean | Gaussian noise | | | Gaussian denoiser [62] | | | JPEG compression | | | JPEG enhancer [62] | | |
|---|---|---|---|---|---|---|---|---|---|---|---|---|---|
| | | $\sigma = 10$ | $\sigma = 20$ | $\sigma = 30$ | $\sigma = 10$ | $\sigma = 20$ | $\sigma = 30$ | Q = 20 | Q = 40 | Q = 80 | Q = 20 | Q = 40 | Q = 80 |
| Baluja [6] | 34.24 | 10.30 | 7.54 | 6.92 | 7.97 | 6.10 | 5.49 | 6.59 | 8.33 | 11.92 | 5.21 | 6.98 | 9.88 |
| ISN [35] | 41.83 | 12.75 | 10.98 | 9.93 | 11.94 | 9.44 | 6.65 | 7.15 | 9.69 | 13.44 | 5.88 | 8.08 | 11.63 |
| HiNet [26] | 42.98 | 12.91 | 11.54 | 10.23 | 11.87 | 9.32 | 6.87 | 7.03 | 9.78 | 13.23 | 5.59 | 8.21 | 11.88 |
| RIIS [68] | 43.78 | 26.03 | 18.89 | 15.85 | 20.89 | 15.97 | 13.92 | 22.03 | 25.41 | 27.02 | 13.88 | 16.74 | 20.13 |
| CRoSS (ours) | 23.79 | 21.89 | 20.19 | 18.77 | 21.39 | 21.24 | 21.02 | 21.74 | 22.74 | 23.51 | 20.60 | 21.22 | 21.19 |

Table 2: PSNR(dB) results of the proposed CRoSS and other methods under different levels of degradations. The proposed CRoSS can achieve superior data fidelity in most settings. The best results are red and the second-best results are blue.

well as species of animals. The concepts of these two prompts can also differ significantly such as the Eiffel Tower and a tree, thereby enhancing the concealment capability and stealthiness of the container images. Meanwhile, the revealed image extracted with "Prompt1" exhibits well fidelity by accurately preserving the semantic information of secret images. Besides prompts, our CRoSS also supports the utilization of various other control conditions as keys, such as depth maps, scribbles, and segmentation maps. As depicted in Fig. 7, our methods can effectively hide and reveal the semantic information of the secret image without significantly compromising the overall visual quality or arousing suspicion. Our CRoSS can also adopt different LoRAs as keys, which is conducive to personalized image steganography.

### 4.4 Property Study#3: Robustness

**Simulation Degradation.** To validate the robustness of our method, we conduct experiments on simulation degradation such as Gaussian noise and JPEG compression. As reported in Tab. 2, our CRoSS performs excellent adaptability to various levels of degradation with minimal performance decrease, while other methods suffer significant drops in fidelity (over 20dB in PSNR). Meanwhile, our method achieves the best PSNR at $\sigma = 20$ and $\sigma = 30$. Furthermore, when we perform nonlinear image enhancement [62] on the degraded container images, all other methods have deteriorations but our CRoSS can still maintain good performance and achieve improvements in the Gaussian noise degradation. Noting that RIIS [68] is trained exclusively on degraded data, but our CRoSS is naturally resistant to various degradations in a zero-shot manner and outperforms RIIS in most scenarios.

**Real-World Degradation.** We further choose two real-world degradations including "WeChat" and "Shoot". Specifically, we send and receive container images via the pipeline of WeChat to implement network transmission. Simultaneously, we utilize the mobile phone to capture the container images on the screen and then simply crop and warp them. Obviously, as shown in Fig. 8, all other methods have completely failed or present severe color distortion subjected to these two extremely complex degradations, yet our method can still reveal the approximate content of the secret images and maintain well semantic consistency, which proves the superiority of our method.

## 5 Conclusion

We propose a coverless image steganography framework named CRoSS (Controllable, Robust, and Secure Image Steganography) based on diffusion models. This framework leverages the unique

properties of diffusion models and demonstrates superior performance in terms of security, controllability, and robustness compared to existing methods. To the best of our knowledge, CRoSS is the first attempt to integrate diffusion models into the field of image steganography. In the future, diffusion-based image steganography techniques will continue to evolve, expanding their capacity and improving fidelity while maintaining their existing advantages.

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
