# CRoSS: Diffusion Model Makes Controllable, Robust and Secure Image Steganography (Supplementary Material)

**Jiwen Yu**[1]     **Xuanyu Zhang**[1]     **Youmin Xu**[1,2]     **Jian Zhang**[1]

[1] Peking University Shenzhen Graduate School     [2] Peng Cheng Laboratory

## A  Data Preparation of Stego260

Regarding the preparation of the Stego260 dataset, we divided it into two stages: image collection stage and prompt preparation stage. Below, we will introduce the details of each stage separately.

**Image Collection Stage.**  In practical applications of image steganography, it is common to hide a single subject in an image, and this is also a problem that our method excels at solving. We selected visually high-quality images from existing datasets to compose our dataset, categorized into three major classes: humans, animals, and general objects. For the human data, we randomly selected 100 images from the publicly available dataset [1]. For the animal data, we randomly chose 100 images with good subjective visual quality from the publicly available dataset [2]. For the general object data, we collected 60 high-quality images of various subjects from the Internet.

**Prompt Preparation Stage.**  We employed two methods to obtain "Prompt1" and "Prompt2": an automated method and a manual method. For the humans and animals categories, we used BLIP [4] to automatically generate "Prompt1", describing the current image. We then input "Prompt1" into ChatGPT to generate the modified "Prompt2". The specific process of generating "Prompt2" is shown in Fig. A.1. However, for the general objects category, we found it challenging to generate accurate textual descriptions of the images using BLIP. Additionally, modifying these descriptions using ChatGPT is difficult as well. Therefore, we adopted a manual approach that involved multiple attempts to select the best "Prompt1" and "Prompt2" for our dataset Stego260.

We present examples from the Stego260 dataset in Fig. A.2, where each example consists of an image and two text prompts. We show images from three categories: humans, animals, and general objects.

## B  More Results for Controllability

To better demonstrate the controllability of the proposed framework, we present more visual results in Fig. B.1 and Fig. B.2. It is obvious that our method is capable of generating container images according to "Prompt2" faithfully, and accurately revealing the secret image based on "Prompt1" targeted at humans, animals or general objects. Furthermore, when utilizing ControlNet [6] with different conditions such as canny edges, depth maps, scribbles and segmentation maps as keys for the image steganography process, our method remains highly controllable and effective.

## C  More Results for Robustness

To verify the robustness of the proposed CRoSS, we impose different degradations on the container images shown in Fig. C.1 and Fig. C.4. "Clean" denotes the clean container images without any degradation. "Resize" denotes the $\times 4$ bicubic downsampling and bicubic upsampling. The noise level of "Gaussian Noise" is set to $20$ and the quality factor of "JPEG" compression is set to $40$. "WeChat" denotes sending and receiving container images via the transmission pipeline of WeChat.

37th Conference on Neural Information Processing Systems (NeurIPS 2023).

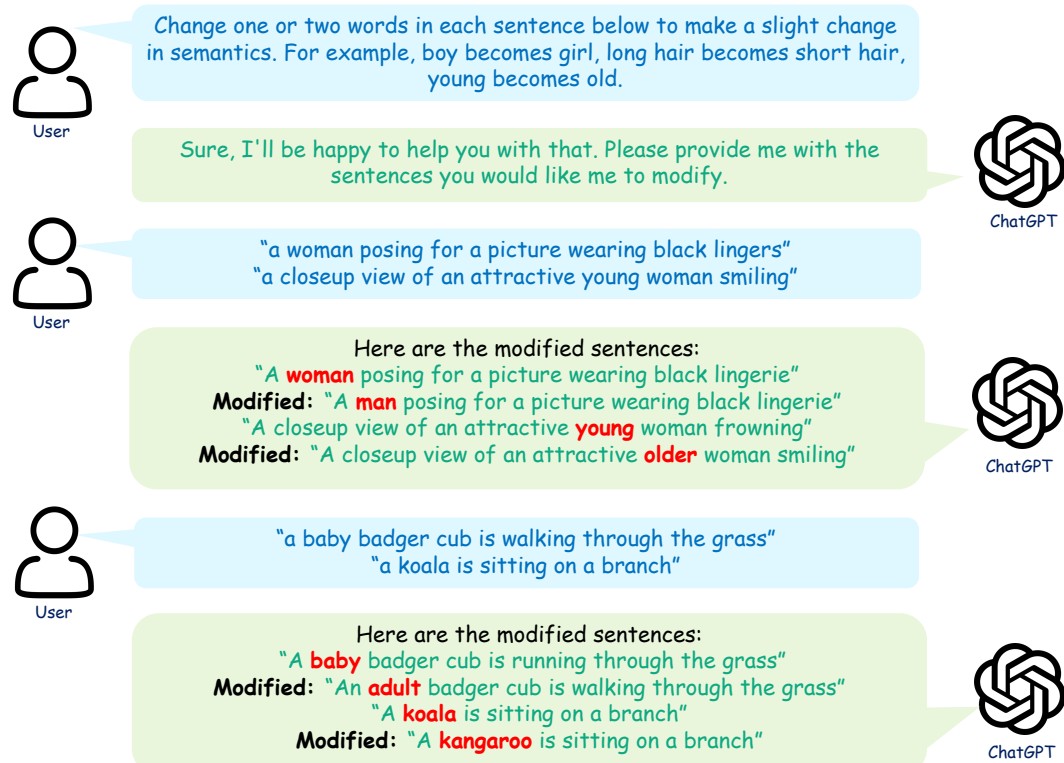

Figure A.1: The demonstration about how to use ChatGPT to get "Prompt2".

"Shoot" denotes capturing the container images via a mobile phone, and simply cropping and resizing it. The parameter $\alpha$ of "Poisson" noise is set to 2 or 4. The probability $p$ of "salt-and-pepper" noise is set to 0.01 or 0.02. For "patch blur", we extract a $256 \times 256$ patch from the center of the image and apply Gaussian blur exclusively to this patch. The blur kernel size is set to 5 and the sigma is set to 2. We compare the proposed CRoSS with two state-of-the-art methods HiNet [3] and RIIS [5].

As illustrated in Fig. C.2, our method can reveal high-quality secret images with fewer artifacts and distortions under the five types of degradation, while RIIS and HiNet fail in most scenarios. Even in some severe degradation like "Shoot", our method can still reveal the general contents of secret images and maintain good semantic consistency, which demonstrates that our approach has great potential for handling complex degradation in real-world applications. Meanwhile, it can be clearly seen from Tab. C and Fig. C.5 that CRoSS maintains robustness against other types of noise such as "Poisson" and "salt-and-pepper" and has an advantage compared to other methods. The experimental results of Fig. C.3 indicate that CRoSS maintains significant robustness against local distortions and exhibits clear advantages compared to other methods.

## D  Limitations

The proposed CRoSS framework offers the possibility of applying diffusion models to image steganography tasks, demonstrating significant advantages in terms of security, controllability, and robustness. However, there are several limitations that await future research and improvement:

- Firstly, although the subjective fidelity of the images revealed by CRoSS is acceptable, there is still a gap in terms of pixel-wise objective fidelity metrics (such as PSNR) compared to cover-based methods. This limitation can be attributed to the fact that the invertibility of zero-shot invertible image translation based on diffusion models is not explicitly guaranteed. Regarding the limitation of the gap in pixel-wise objective fidelity metrics, which is primarily attributed to the training-free nature of DDIM Inversion, a potential solution could involve training diffusion models specifically for image steganography tasks to improve objective fidelity.

- Secondly, to ensure the zero-shot invertibility of image translation based on diffusion models, there is a trade-off that sacrifices the editing capability. Therefore, CRoSS excels at modifying single subjects within the secret image but struggles to modify the global contents of the secret image. This potential limitation of CRoSS arises in scenarios where it is necessary to hide the global contents of the secret image. Regarding the limitation of sacrificing editing capability to ensure invertibility, which is mainly attributed to the unsatisfactory generation capabilities of Stable Diffusion (with guidance scale $= 1$), a potential strategy involves keeping up with the latest advancements in the diffusion model field and replacing the base model in CRoSS in a plug-and-play way to alleviate this issue.

- Thirdly, CRoSS currently supports hiding only one secret image within a single container image. However, in practical applications of image steganography, it is desirable to hide as many secret images as possible within a single container image. Exploring new methods to increase the capacity of diffusion-based image steganography techniques is of great value. Regarding the limitation of hiding only one secret image within a single container image, this is primarily due to the current design of diffusion models being Single Input Single Output (SISO). A potential solution could involve designing Multiple Input Multiple Output (MIMO) diffusion models specifically for high-capacity image steganography.

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

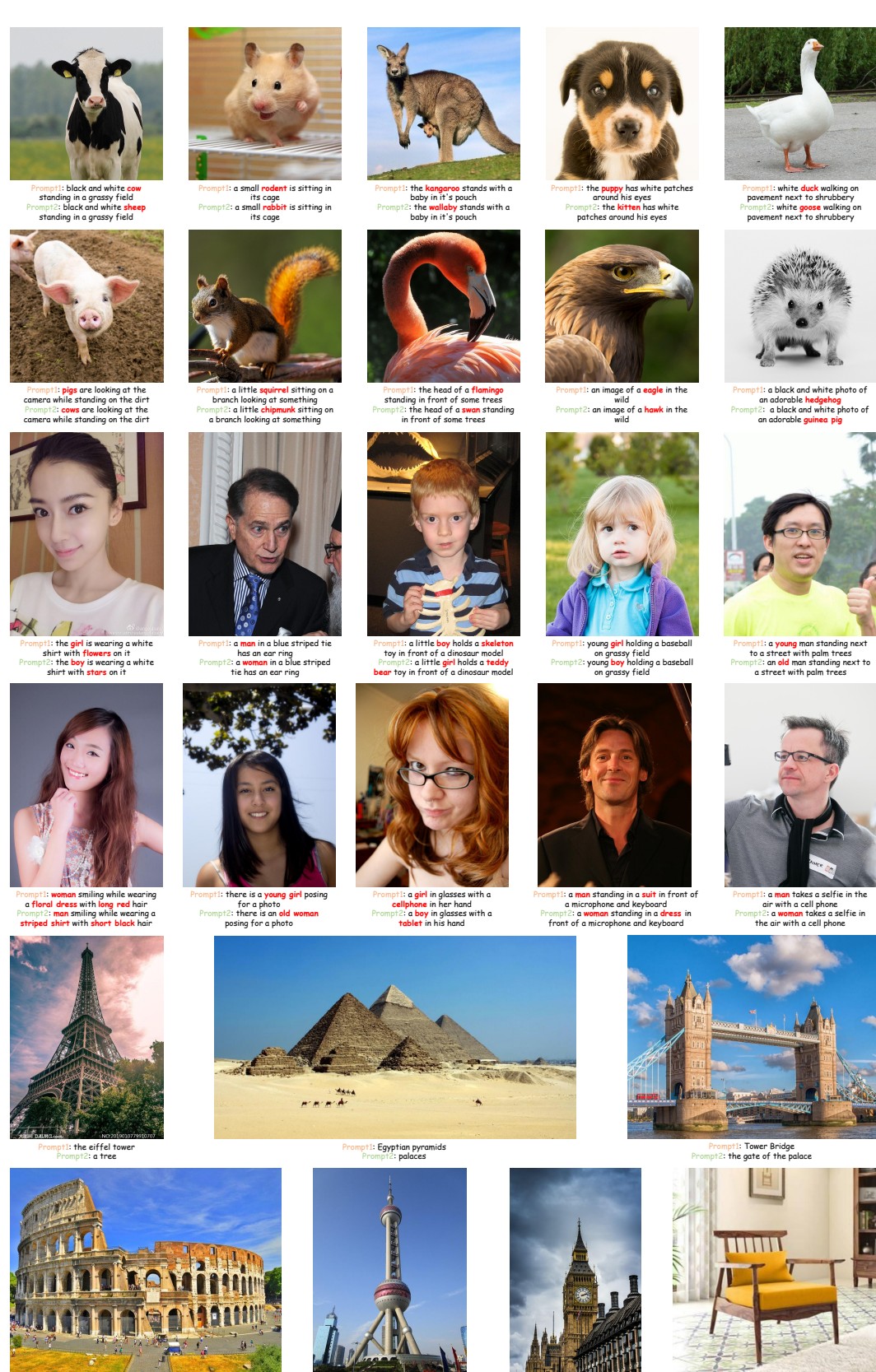

Figure A.2: We showcase examples from three categories: humans, animals, and general objects. Each example consists of an image and two text prompts. "Prompt1" is used to describe the content of the image, while "Prompt2" is intended to modify the content of the image. If "Prompt1" and "Prompt2" are long sentences, we emphasize their differences using **bold red font**.

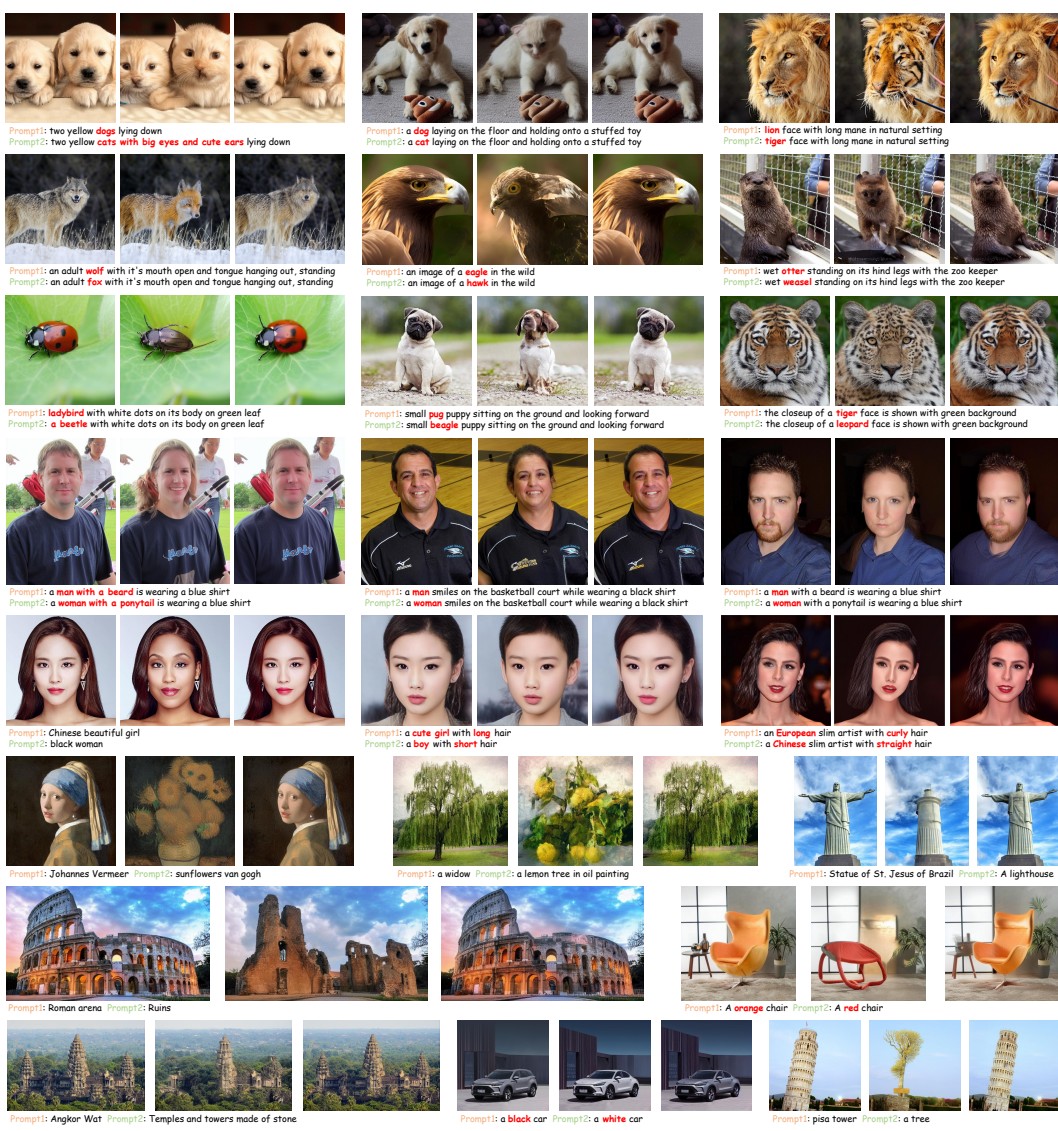

Figure B.1: Visual results of the proposed CRoSS controlled by different prompts on three categories namely humans, animals and general objects. The proposed CRoSS can produce realistic image details and fine-grained textures guided by the "Prompt2" and accurately reveal them. If "Prompt1" and "Prompt2" are long sentences, we emphasize their differences using **bold red font**.

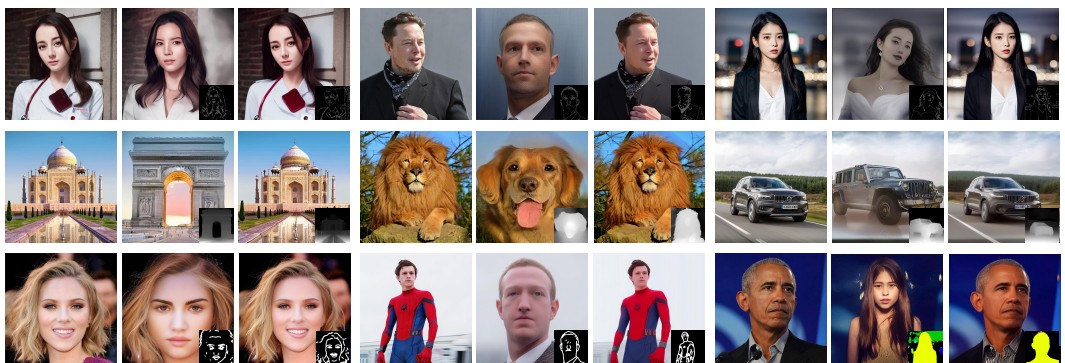

Figure B.2: Visual results of the proposed CRoSS controlled by different conditions like canny edges, depth maps, scribbles, and segmentation maps. The condition maps are placed in the right corner of container images and revealed images.

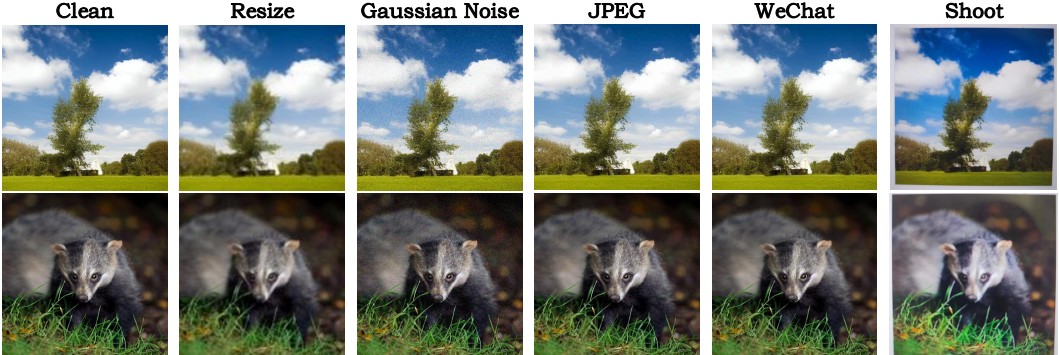

Figure C.1: Visualization of different degradations used to corrupt the container images of CRoSS. "Clean" denotes the clean container images without any degradation. **Zoom in for best view.**

Figure C.2: Revealed images of CRoSS and other methods under different degradation.

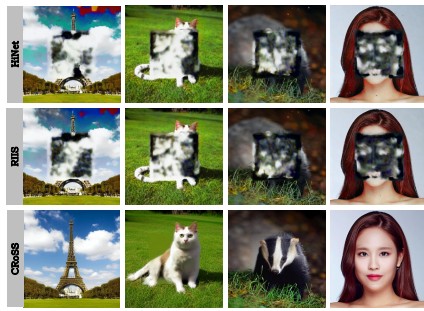

Figure C.3: Visualization of the revealed images of CRoSS and other methods under "patch blur" degradation. **Zoom in for best view.**

Table C.1: PSNR(dB) results of the proposed CRoSS and other methods under different levels of degradations. The proposed CRoSS can achieve superior data fidelity in most settings. The best results are red and the second-best results are blue.

| Methods | Poisson | | Salt-and-pepper | |
|---|---|---|---|---|
| | $\alpha = 2$ | $\alpha = 4$ | $p = 0.01$ | $p = 0.02$ |
| Baluja | 8.94 | 10.23 | 12.45 | 9.82 |
| ISN | 13.45 | 14.53 | 16.32 | 15.49 |
| HiNet | 14.32 | 16.28 | 16.65 | 14.89 |
| RIIS | 19.76 | 21.16 | 20.86 | 19.14 |
| CRoSS (ours) | 20.11 | 23.26 | 22.09 | 19.20 |

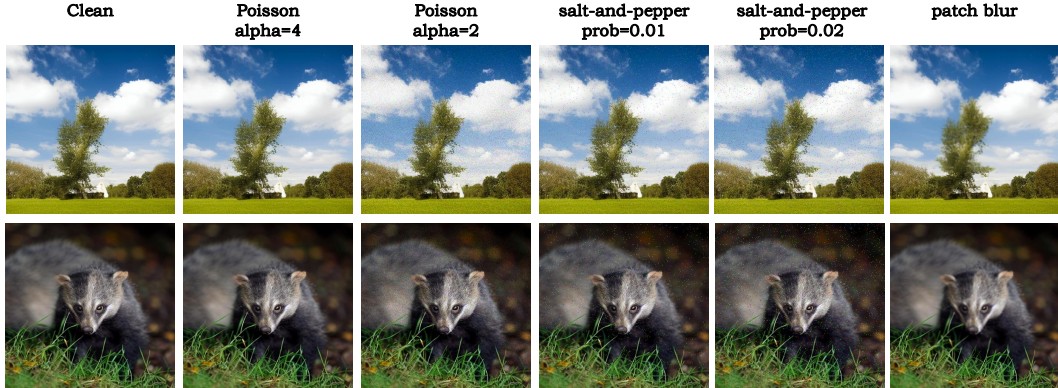

Figure C.4: Visualization of different degradation used to corrupt the container images of CRoSS. "Clean" denotes the clean container images without any degradation. **Zoom in for best view.**

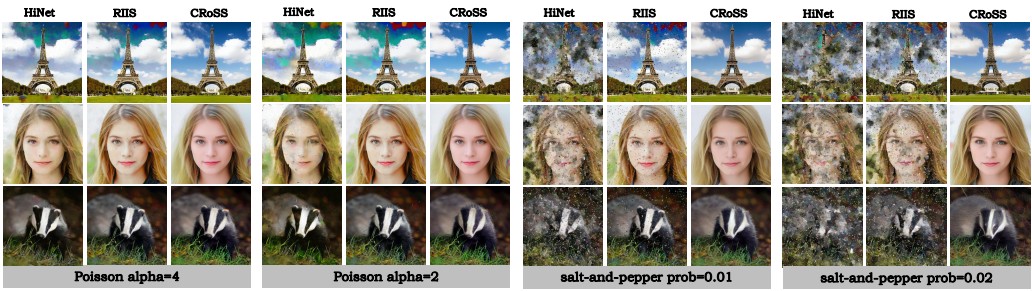

Figure C.5: Revealed images of CRoSS and other methods under different degradation. **Zoom in for best view.**