# OpenReview forum: "CRoSS: Diffusion Model Makes Controllable, Robust and Secure Image Steganography"
_NeurIPS.cc/2023/Conference — NeurIPS 2023 poster_

### Official Review · Reviewer_1iwK · 2023-06-14

**Soundness:** 3 good
**Presentation:** 2 fair
**Contribution:** 2 fair
**Rating:** 4
**Confidence:** 4

**Summary:**

This paper presents CRoSS, a novel image steganography framework leveraging text-driven diffusion models. It offers improved security, robustness, and controllability compared to cover-based methods.

**Strengths:**

CRoSS is the first work to introduce diffusion models to image steganography and achieves these benefits without requiring additional training.

**Weaknesses:**

- Equation 3 should be written in detail. Is it the same as those in Section H of appendix in guided diffusion [A]？

- The collected dataset only contains 260 images, rendering it not convincing in the conclusions from the presented quantitative experiments.  Besides, the secret and stego image pairs much resemble either in appearance or in concept, e.g., cat versus tiger, or man versus woman. As a result, there is a restriction of secret-stego correlation that does not exist in the compared previous works.

- Stable diffusion [B] utilizes VAE to reconstruct images from the latent space, leading to inherent errors. As a result, the performance of CRoSS is constrained by this factor, resulting in significant errors. In hiding tasks, the extraction accuracy is a crucial metric. But the PSNR between the revealed images and the corresponding secret images is merely 23.79dB, or even less when there is distortion, suggesting that the method can be less applicable in the real world.

- For each instance of covert transmission using the proposed method, the recipient must know the exact prompt or condition, e.g., word, sketch, or even parameters, used during hiding, which might be costy and inpractical.

- As stated in [C], setting the guidance scale to 1 (used in the classifier-free control algorithm) may compromise the image editability. But in this paper, it is indeed set to 1, which could potentially limit the range of secret image selection.


[A] Dhariwal P, Nichol A. Diffusion models beat gans on image synthesis[J]. Advances in Neural Information Processing Systems, 2021, 34: 8780-8794.

[B] Rombach R, Blattmann A, Lorenz D, et al. High-resolution image synthesis with latent diffusion models[C]//Proceedings of the IEEE/CVF Conference on Computer Vision and Pattern Recognition. 2022: 10684-10695.

[C] Mokady R, Hertz A, Aberman K, et al. Null-text inversion for editing real images using guided diffusion models[C]//Proceedings of the IEEE/CVF Conference on Computer Vision and Pattern Recognition. 2023: 6038-6047.


**Questions:**

- Please conduct large-scale quantitative experiments, author may refer to the the settings in InstructPix2Pix [A].
- Please conduct experiments using prompts that show low relation with the secret images.
- More distortions shall be considered to verify general robustness.

[A] Brooks T, Holynski A, Efros A A. Instructpix2pix: Learning to follow image editing instructions[C]//Proceedings of the IEEE/CVF Conference on Computer Vision and Pattern Recognition. 2023: 18392-18402.

---

> ### Author Rebuttal · Authors · 2023-08-05
>
> Thank you for your constructive comments! If there are any remaining questions that have not been adequately addressed, please feel free to continue the discussion with us.
>
> > ***We earnestly request you to reconsider your assessment of our work, taking into consideration different aspects.***
>
> - We understand that your concerns about our work primarily center around objective fidelity and practical applicability. From our perspective, CRoSS holds distinct advantages in terms of $\textbf{\textcolor{red}{security and robustness}}$, which significantly enhance its $\textbf{\textcolor{red}{practical applicability}}$.
> - The introduction of diffusion models for the $\textbf{\textcolor{red}{first}}$ time introduces new directions and tools in the field of image steganography, $\textbf{\textcolor{red}{inspiring}}$ future research.
>
> > ***Weakness #1: The details of Equation 3 and its relationship with guided diffusion.***
>
> - Equation 3 describes the sampling process of DDIM [1], while the appendix H in guided diffusion [2] discusses sampling process of conditional diffusion models. $\textbf{\textcolor{red}{These are distinct concepts and not closely related}}$.
> - As the specific details of the DDIM sampling process are not a central focus of our work, we employed the concise form presented in Equation 3. This type of representation is commonly used in many relevant literatures due to its brevity.
> - The details of Equation 3 involve iteratively executing the single-step sampling formula (Equation 2) using a FOR loop, sampling from $\mathbf{x}_T$ to $\mathbf{x}_0$ .
>
> > ***Weakness #2 & Question #1: Insufficient size of Stego260 and similarity between secret and container images.***
>
> - On one hand, the dataset size of Stego260 is not considered small. For instance, some previous works such as RIIS (CVPR 2022) [3], only conducted evaluation on just 100 images from DIV2K.
> - On the other hand, creating a large-scale dataset for image steganography with text prompt labels (similar to Instruct-Pix2Pix [4]) is both costly and not a primary focus of our study. It might be more appropriate to leave this issue for future work.
> - In our experimental setting, there is indeed an issue of similarity between secret images and container images. This is primarily attributed to the limited editing capability of Stable Diffusion (with guidance scale = 1). However, this aspect is **independent** of our proposed CRoSS framework. If in the future there are more powerful diffusion-based editors available, they could be seamlessly integrated into CRoSS, potentially alleviating the concern you raised.
>
> > ***Weakness #3: Concerns about objective fidelity and practicality.***
>
> - The fidelity of our results is subjectively acceptable in terms of visual perception, and the quality does not reach the level of "significant errors" as you mentioned. Our emphasis is not on pixel-level objective fidelity metrics such as PSNR, but rather on semantic fidelity.
> - It's important to emphasize that evaluating an image steganography algorithm requires a $\textbf{\textcolor{red}{comprehensive consideration}}$ of three aspects: fidelity, security, and robustness. CRoSS exhibits significant advantages in terms of security and robustness, and considering that its fidelity is subjectively acceptable, CRoSS should indeed have $\textbf{\textcolor{red}{great practical advantages}}$ in real-world applications.
>
> > ***Weakness #4: Practicality of the private and public keys such as prompts.***
>
> - In most cases, we use text prompts as the key. In this scenario, we have the following reasons to demonstrate the practicality of using text prompts:
>     - (1) The transmission and storage costs of text prompts are **highly efficient**.
>     - (2) During the reveal process, it's not necessary for each word to be exact; rather, a **general semantic correctness** is sufficient.
> - For other types of keys, they are extensions we introduced to showcase the diversity and extensibility of the CRoSS framework.
>
> > ***Weakness #5 & Question #2: The editability is limited with 1 as guidance scale.***
>
> - To ensure invertibility, we opted for Stable Diffusion (with guidance scale = 1), which indeed has limitations in terms of generation and editing capabilities. Therefore, it may not perform well in cases where the correlation between the given prompt and secret image is low.
> - However, this limitation is inherent to Stable Diffusion itself. The focus of the CRoSS work is to introduce a novel framework, which is **independent** of the specific diffusion model chosen. If in the future, more powerful diffusion-based editors (such as the latest SDXL [5]) become available, they can be readily integrated as a plug-and-play replacement for Stable Diffusion to achieve improved outcomes.
>
> > ***Question #3: More distortions should be considered to verify general robustness.***
>
> - Regarding distortion types, we have conducted experiments involving five different types: **Resize, Gaussian noise, JPEG compression, WeChat, and Shoot**. Particularly, the WeChat and Shoot categories include complex real-world degradations, such as information loss due to **image compression, color distortion, and moiré patterns**.
> - Following the advice from Reviewer mtD2, we have also added supplementary experiments in the supplementary rebuttal PDF file, including **Poisson noise, salt-and-pepper noise and blurring-a-patch distortion**.
> - In conclusion, the number of distortion types included in the robustness experiments should be $\textbf{\textcolor{red}{sufficient}}$ to clearly demonstrate the significant advantages of CRoSS in terms of robustness.
>
> > ***References:***
>
> [1] Denoising Diffusion Implicit Models. ICLR 2021.
>
> [2] Diffusion Models Beat Gans on Image Synthesis. NeurIPS 2021.
>
> [3] Robust Invertible Image Steganography. CVPR 2022.
>
> [4] InstructPix2Pix: Learning to Follow Image Editing Instructions. CVPR 2023.
>
> [5] https://stablediffusionxl.com/

---

> > ### Comment · Reviewer_1iwK · 2023-08-19
> > **Response**
> >
> > I have read the rebuttal and some concerns such as Weakness #2, have been addressed.
> > However, I still maintain some issues as follows:
> > + Weakness #1: Actually, Appendix H in Guided Diffusion can be viewed as a detailed representation of the DDIM-sampled ODE equation. And diffusion models can be applied with diverse ODEs, such as EDM[1]. Therefore, a detailed description is warranted.
> > + Weakness #3#5: The objective of a steganographic system is to accurately convey secret messages. Accuracy on extraction is an indispensable metric. Moreover, cover and stego images often share some visual elements like backgrounds, which can compromise security to a certain extent. Setting the guidance scale to 1, in my opinion, restricts the breadth of real-world application scenarios.
> > + Weakness #4: The authors mentioned that during the reveal process, it's possible to guess the prompt key. If feasible, it would be helpful to showcase relevant examples and demonstrate the extraction results.
> >
> > Thanks for the authors' response. Looking forward to further discussion.
> >
> > **Reference**:
> >
> > [1] Elucidating the design space of diffusion-based generative models. NIPS 2022.

---

> > > ### Author Response · Authors · 2023-08-19
> > > **Response to Reviewer 1iwK's Concerns (part 1)**
> > >
> > > Thank you for your timely response and active engagement in the discussion! Below is our latest response:
> > >
> > > > ***Weakness #1: The details of Equation 3 and its relationship with appendix H of guided diffusion***
> > >
> > > We have carefully read Appendix H of the guided diffusion paper [1] (titled "Conditional Diffusion Process") and would like to provide some clarifications regarding your comments:
> > > - Appendix H of [1] provides an overview of the theory behind the **conditional diffusion model**. The conditional diffusion model can be represented by $q(x_{t}|x_{t+1}, y)$. Using Bayes' theorem, it can be further broken into two components: $q(x_{t}|x_{t+1})$ representing the unconditional diffusion model and $q(y|x_{t})$ representing conditional guidance. For more detailed insights, please refer to Equation 55~61 in Appendix H of [1] along with the subsequent analysis. It's important to emphasize that the theory of the conditional diffusion model discussed in Appendix H of [1] is $\textcolor{red}{\textbf{independent of specific sampling algorithms}}$, whether it's DDIM or DDPM, ODE or SDE.
> > > - Our Equation 3 describes the general DDIM sampling process, which is applicable to both conditional and unconditional diffusion models. In summary, there is a **noticeable distinction** between Equation 3 and Appendix H of [1].
> > >
> > > However, we also find your suggestion to provide more detailed description for Equation 3 constructive. We will incorporate your suggestion into the revised version.
> > >
> > > > ***Weakness #3&#5: Issues of accuracy on extraction and the setting of "guidance scale=1"***
> > > - Regarding the issue of fidelity, or more specifically, the **accuracy on extraction**, our clarification is as follows:
> > >     - Firstly, we agree with your perspective that fidelity is a crucial aspect of image steganography algorithms. However, achieving higher PSNR solely **under ideal conditions** falls short of practical significance. In real-world scenarios, container images often carry complex real-world distortions. Investigating fidelity under such conditions is more practical and significant. However, $\textcolor{red}{\textbf{this point has been overlooked in previous works}}$, which is precisely where CRoSS holds a unique advantage. CRoSS demonstrates better fidelity under various distortion conditions, showcasing enhanced robustness.
> > >     - Additionally, we believe that pixel-wise evaluation metrics like PSNR can be misleading. A significant amount of research has concentrated on boosting PSNR **under ideal conditions**, disregarding real-world application needs. While our CRoSS may not achieve the same level of **objective fidelity** as compared to previous methods, its **subjective fidelity** in terms of visual perception is indeed satisfactory.
> > >     - Lastly, $\textcolor{red}{\textbf{there is substantial room for improvement in CRoSS's objective fidelity.}}$ There are two possible approaches: (1) Not using latent diffusion models like Stable Diffusion but employing image diffusion models such as Imagen [2]. (2) Introducing strategies that require training. It's worth noting that CRoSS currently operates **without requiring training**. However, by maintaining the overall framework and simultaneously incorporating dedicated training for image steganography tasks, it is possible to mitigate its objective fidelity disadvantage while preserving its advantages in robustness and security. This would be a valuable topic for the future research.
> > > - Regarding the issue of background similarity and the setting of **"guidance scale=1"**, our clarification is as follows:
> > >     - Currently, in order to ensure subjective fidelity, we have adopted the Stable Diffusion with a guidance scale of 1. This has indeed led to certain issues such as background similarity. What we want to emphasize here is that these issues $\textcolor{red}{\textbf{do not stem from the CRoSS framework itself}}$, but rather from the specific choice of the diffusion model used.
> > >     - Because the editing capability of Stable Diffusion with a guidance scale of 1 is limited, we adopted experimental settings such as using similar backgrounds. However, the generative capability of diffusion models is **continuously evolving**. For instance, recent advancements like the upgraded version of Stable Diffusion (SDXL) are further pushing the boundaries of diffusion model capabilities. In the future, we hope to leverage **more powerful** classifier-free diffusion models with a guidance scale of 1, or even non-classifier-free diffusion models, to achieve **stronger editing capabilities**.
> > >     - This way, we can enhance the CRoSS framework in a **plug-and-play manner**. CRoSS, as the first effort to integrate diffusion models into image steganography, introduces a new framework, showcasing pioneering value.
> > >
> > > > ***References***
> > >
> > > [1] Diffusion Models Beat Gans on Image Synthesis. NeurIPS 2021.
> > >
> > > [2] Photorealistic Text-to-Image Diffusion Models with Deep Language Understanding. NeurIPS 2022.

---

> > > ### Author Response · Authors · 2023-08-19
> > > **Response to Reviewer 1iwK's Concerns (part 2)**
> > >
> > > > ***Weakness #4: Relevant examples when the receiver guesses the prompt***
> > >
> > > - In the main paper, in **Figure 4** on the far left (Scenario #1), we illustrate potential candidate revealed results that the receiver might generate when trying to guess the prompt. As can be observed, for the receiver, it is not feasible to deduce the correct answer through random guessing of the prompt, as each candidate revealed image appears authentic.
> > > - We will incorporate your instructive suggestion and provide more examples in the revised version.
> > >
> > > We hope that our responses have addressed all of your concerns. We sincerely appreciate your continued engagement!

---

> > > ### Author Response · Authors · 2023-08-21
> > > **Looking forward to further discussions with the Reviewer 1iwK**
> > >
> > > Dear Reviewer 1iwK,
> > >
> > > Thank you for your continued engagement in the discussion!
> > >
> > > We apologize for any inconvenience caused by reaching out again. However, today marks the last day of the discussion phase, and we have provided detailed analysis and clarifications in response to your latest concerns. We would like to further discuss with you to ensure that your concerns are addressed.
> > >
> > > Looking forward to our continued discussion!
> > >
> > > Best regards,
> > >
> > > Authors

---

> ### Author Response · Authors · 2023-08-18
> **Looking forward to discussions with the Reviewer 1iwK**
>
> Dear Reviewer 1iwK,
>
> We appreciate your previous review time and constructive comments. In response to your concerns, we have provided explanations, clarifications, and additional experimental results in the supplementary rebuttal PDF file.
>
> We would like to know if your concerns have been adequately addressed. If you have further questions or comments, we would be more than willing to address and discuss them.
>
> Thank you for your efforts!
>
> Best regards,
>
> Authors

---

### Official Review · Reviewer_g14p · 2023-06-26

**Soundness:** 2 fair
**Presentation:** 1 poor
**Contribution:** 1 poor
**Rating:** 3
**Confidence:** 5

**Summary:**

This paper addresses coverless image steganography by taking the prompt as the guidance to generate stego images using Stable Diffusion. It shows better controllability with language-driven model, better robustness and security with stronger generation power of diffusion probabilistic model. Experimental results show better performance than SOTAs.

**Strengths:**

+) The first coverless stegnography that uses pretrained diffusion model for image steganography;
+) Prompt as control of steganography

**Weaknesses:**

-) Novelty is somewhat limited. It is obvious that using fixed sampling could always generate the same image using a pretrained diffusion model. I'd like to see contributions in terms of network structures and insights that drives coverless steganography in general.
-) Not clear about how the public and private keys are used in the model.
-) Missing experiments on popular datasets such as  BOSSbase and BOWS2.
-) Another major problem is RIIS performance better than CRoSS in some cases. Please explain them in details. It is not proper that RIIS is trained on degraded data compared to CRoSS, as both can be set under the same setting.


**Questions:**

+) In Figure 8, what does "Shoot" mean?
+) Typo error: In Algorithm1, L2, hided -> hidden
+) Does CRoSS have any limitations?

**Limitations:**

Not mentions. I'd like to hear about any potential limitations.

---

> ### Author Rebuttal · Authors · 2023-08-05
>
> Thank you for your constructive comments! If there are any additional comments to be added, please continue the discussion with us.
>
> > ***Weakness #1: The novelty is limited.***
>
> - Our contributions are primarily demonstrated in the following aspects. $\textbf{\textcolor{red}{These major contributions firmly establish the novelty of our work}}$:
>     - (1) We are the $\textbf{\textcolor{red}{first}}$ to apply diffusion models to image steganography.
>     - (2) Our approach exhibits $\textbf{\textcolor{red}{significant advantages}}$ in terms of security and robustness.
>     - (3) CRoSS, as a novel image steganography framework, can provide $\textbf{\textcolor{red}{valuable insights}}$ for future research in the field of image steganography.
> - We are uncertain why you mentioned the point about "fixed sampling generating the same image," as it appears unrelated to our work. Could you please provide further clarification or elaboration on this matter?
> - Our work does not focus on introducing a new model structure; instead, our objective is to present a novel image steganography framework. This framework inherently offers a wealth of insights to inspire future research.
>
> > ***Weakness #2 & Question #1 & Question #3: Some missing contents (but we have already provided them)***
>
> - **(Weakness #2)** We have provided detailed explanations about private and public keys in two separate paragraphs in $\textbf{\textcolor{red}{Section 3.3 (Lines 206-212, Lines 231-237)}}$. Could you please specify the specific points that you find unclear so that we can address them more precisely?
> - **(Question #1)** We have already explained the meaning of "Shoot" in $\textbf{\textcolor{red}{Section 4.4 (Lines 305-306)}}$. The meaning of "Shoot" is: we utilize the mobile phone to capture the container images on the screen and then simply crop and warp them.
> - **(Question #3)** We have thoroughly discussed the limitations of CRoSS in $\textbf{\textcolor{red}{Section D}}$ of the supplementary materials.
>
> > ***Weakness #3: Missing experiments on other datasets.***
>
> - The two datasets you mentioned, BOSSbase and BOWS2, both consist of 10,000 grayscale images with dimensions of 512x512. As far as we are aware, many studies related to image steganography have not conducted experiments on them, such as RIIS (CVPR 2022) [1], HiNet (ICCV 2021) [2], and ISN (CVPR 2021) [3].
> - Moreover, these two datasets $\textbf{\textcolor{red}{do not align with our experimental setting}}$. For evaluation, CRoSS requires labeled text prompts for secret images. BOSSbase and BOWS2 do not meet this specific requirement. Therefore, we gathered and labeled the Stego260 dataset ourselves for evaluation, and this dataset can also facilitate future related research.
>
> > ***Weakness #4: Comparison between RIIS and CRoSS***
>
> - Please note that RIIS [1] is a method that requires training specific to degradation types, whereas CRoSS $\textbf{\textcolor{red}{does not require training}}$. Therefore, RIIS and CRoSS cannot share the same setting and it's not surprising that RIIS performs better in scenarios with no degradation or some synthetic degradation, as opposed to CRoSS.
> - However, CRoSS exhibits $\textbf{\textcolor{red}{significant advantages}}$ in cases involving real-world degradations such as WeChat and Shoot. Furthermore, it's not possible for RIIS to train on all types of degradations, including real-world degradations. Therefore, CRoSS holds clear $\textbf{\textcolor{red}{practical advantages}}$ over RIIS, and such a comparison is appropriate.
>
> > ***Question #2: A typo***
>
> Thank you for pointing out this typo. We will correct it in the revised version.
>
> > ***References***
>
> [1] Robust Invertible Image Steganography. CVPR 2022.
>
> [2] HiNet: Deep Image Hiding by Invertible Network. ICCV 2021.
>
> [3] Large-capacity Image Steganography based on Invertible Neural Networks. CVPR 2021.

---

> > ### Comment · Reviewer_g14p · 2023-08-11
> > **Response**
> >
> > I appreciate the responses from authors. However, I have concerns regarding:
> > 1. Steganalysis. A well steganography method is capable of defending various steganalysis methods. The popular SRNet [1] and SigstegNet [2] are needed to validate performances of the steganography method. I believe SiastegNet is an improved version of SID.
> > 2. I believe CRoSS and RIIS can share the same setting when both of them are trained on the same setting. Otherwise you could not compare the two. According to the numeric results, I'm afraid it's no better than RIIS.
> >
> > [1] Mehdi Boroumand, et al, Deep Residual Network for Steganalysis of Digital Images, http://www.ws.binghamton.edu/fridrich/research/SRNet.pdf
> > [2] Weike You et al, A Siamese CNN for Image Steganalysis, TIFS

---

> > > ### Author Response · Authors · 2023-08-12
> > > **Response to Reviewer g14p's Concerns**
> > >
> > > Thank you for your prompt response. Below is our response for your concerns:
> > >
> > > > ***Experiments involving various steganalysis methods***
> > >
> > > - Firstly, our paper $\textbf{\textcolor{red}{already}}$ includes experiments involving various steganalysis methods, as shown in $\textbf{\textcolor{red}{Figure 5 and Table 1}}$ of the main paper. These methods include **SID** [1], **StegExpose** [2], **XuNet** [3], **YedroudjNet** [4], and **KeNet** [5]. $\textbf{\textcolor{red}{It's worth noting that KeNet [5] is the same as the method SigStegNet you mentioned.}}$ The authors of [5] chose the name **"KeNet"** for **SigStegNet**, and you can find more details about this naming choice in their GitHub link [6].
> > > - Secondly, we have included experiments involving detection accuracy of **SRNet** [7] as you mentioned. The closer the detection accuracy is to 50%, the better the security. The results of these experiments are as follows and demonstrate the superiority of CRoSS over other methods in terms of security:
> > >
> > > | #Leaked Samples | 60 | 120 | 180 |
> > > | ----- | ----- | ----- | ----- |
> > > | ISN | 57.5% | 67% | 66.75% |
> > > | Baluja | 55.75% | 65% | 71.5% |
> > > | HiNet | 58.67% | 63.5% | 68.33% |
> > > | RIIS | 54.25% | 61.67% | 59.5% |
> > > | **CRoSS** | **54%** | **59.5%** | **57.25%** |
> > >
> > > - In conclusion, we believe that the number of steganalysis methods involved in the experiments is $\textbf{\textcolor{red}{sufficient}}$ to demonstrate the security advantages of CRoSS.
> > >
> > > > ***Experiments with the same settings for both RIIS and CRoSS***
> > >
> > > - We need to emphasize again that $\textbf{\textcolor{red}{CRoSS does not require training for specific distortion types.}}$ In order to make CRoSS and RIIS share the same setting, we compared the robustness differences between RIIS and CRoSS, where RIIS **was not trained under any distortion types.** The PSNR results are shown below, and CRoSS outperforms RIIS (**"GN"** refers to Gaussian noise):
> > >
> > > | Methods | GN ($\sigma=10$) | GN ($\sigma=20$) | GN ($\sigma=30$) | JPEG ($Q=20$) | JPEG ($Q=40$) | JPEG ($Q=80$) |
> > > | --- | --- | --- | --- | --- | --- | --- |
> > > | RIIS | 13.78 | 12.73 | 11.09 | 7.48 | 10.05 | 13.82 |
> > > | **CRoSS** | **21.89** | **20.19** | **18.77** | **21.74** | **22.74** | **23.51** |
> > >
> > > - Furthermore, based on the experimental results we have already provided, CRoSS $\textbf{\textcolor{red}{undoubtedly outperforms}}$ RIIS in terms of robustness and security. These results include:
> > >     - Most results in $\textbf{\textcolor{red}{Figure 5 and Table 1}}$ of the main paper (security experiments).
> > >     - $\textbf{\textcolor{red}{Figure 8}}$ of the main paper (subjective results of robustness experiments).
> > >     - Most results in $\textbf{\textcolor{red}{Table 2}}$ of the main paper (objective results of robustness experiments).
> > >     - $\textbf{\textcolor{red}{Figure B.4}}$ in the supplementary material (more subjective results of robustness experiments).
> > >     - $\textbf{\textcolor{red}{Figure 1, Figure 3 and Table 1}}$ in supplementary rebuttal PDF file (more subjective and objective results of robustness experiments).
> > >
> > > > ***References***
> > >
> > > [1] Steganalyzing images of arbitrary size with CNNs. Electronic Imaging 2018.
> > >
> > > [2] Stegexpose-A tool for detecting LSB steganography. ArXiv 2014.
> > >
> > > [3] Structural design of convolutional neural networks for steganalysis. IEEE Signal Processing Letters 2016.
> > >
> > > [4] Yedroudj-net: An efficient CNN for spatial steganalysis. ICASSP 2018.
> > >
> > > [5] A Siamese CNN for image steganalysis. TIFS 2020.
> > >
> > > [6] https://github.com/SiaStg/SiaStegNet#quickstart
> > >
> > > [7] Deep residual network for steganalysis of digital images. TIFS 2018.

---

> > > > ### Comment · Reviewer_g14p · 2023-08-18
> > > > **Response**
> > > >
> > > > I read the rebuttal. Authors have address my first question well, however, I still maintain my concern that the comparison between RIIS and CROSS is not fair.
> > > > Besides, I noticed that Reviewer g14p has mentioned that extraction accuracy for hiding task is quite important. The PSNR of the revealed image hardly reaches 40dB, which is not good.
> > > > Therefore, I maintain my initial rating.

---

> > > > > ### Author Response · Authors · 2023-08-19
> > > > > **Response to Reviewer g14p's Concerns**
> > > > >
> > > > > Thank you for your response. We need to emphasize again that:
> > > > > - Evaluating an image steganography algorithm requires a $\textcolor{red}{\textbf{comprehensive assessment of fidelity, security, and robustness.}}$ Dismissing CRoSS **solely** based on fidelity **under ideal conditions** is one-sided and unreasonable.
> > > > > - Under ideal conditions, RIIS, as a method that **requires training**, indeed exhibits better fidelity than CRoSS, which **doesn't require training**. However, in practical scenarios, distortions inevitably appear on container images, and image steganography algorithms face the risk of detection. At this point, the performance in terms of **security and robustness** becomes particularly crucial. CRoSS holds a distinct advantage in these aspects, which is a valuable insight we gained by introducing diffusion models into image steganography. $\textcolor{red}{\textbf{Comparing CRoSS and RIIS in real-world application scenarios is reasonable}}$, and we fail to comprehend why you consider this comparison unfair.

---

> > > ### Author Response · Authors · 2023-08-18
> > > **Looking forward to further discussions with the Reviewer g14p**
> > >
> > > Dear Reviewer g14p,
> > >
> > > We appreciate your previous review and the prompt response. We have responded to your latest concerns and supplemented the experiments on steganalysis methods and the comparison between RIIS and CRoSS.
> > >
> > > We would like to further discuss whether your concerns have been addressed. If there are any aspects of our work that are still unclear to you, please let us know.
> > >
> > > Thank you for your continued engagement!
> > >
> > > Best regards,
> > >
> > > Authors

---

### Official Review · Reviewer_Nci3 · 2023-07-06

**Soundness:** 3 good
**Presentation:** 3 good
**Contribution:** 3 good
**Rating:** 7
**Confidence:** 4

**Summary:**

This paper introduces diffusion models to the field of image steganography. It argues the significant advantages in controllability, robustness, and security compared to cover-based image steganography methods. It utilized the power of Stable Diffusion to translate between two images without training. This paper collects a benchmark and shows experiments on it.

**Strengths:**

1. This paper proposes a novel idea to utilize the diffusion Model to do image steganography. I think this is reasonable because pretrained diffusion models can invert an image to noise or translate between images without training.
2. The presentation is clear such as Fig.2 and Fig.3.
3. The qualitative results seem good as shown in Fig. 6.
4. This paper collects a benchmark containing 260 images, which may be useful for the community.
5. Analysis is in detail, including discussion from Sec 4.2 to Sec 4.4.

**Weaknesses:**

1. According to null-text inversion [1], DDIM is not enough to invert a real image perfectly. In this case, how to deal with the artifacts in your method? I believe low-level artifacts are fatal for image steganography.
2. Most secrets and containers share a similar background. Does it mean we need a good image editing tool when using this method?
3. It is difficult to generate high-quality human faces with Stable Diffusion. Inverting human faces is also not stable. Why do the qualitative results in Fig. 6 seem good?


[1] Null-text Inversion for Editing Real Images using Guided Diffusion Models

**Questions:**

1. Why Secret and Container are required to share a similar background?

**Limitations:**

Please refer to the "weakness".

---

> ### Author Rebuttal · Authors · 2023-08-05
>
> Thank you for your constructive comments! We hope that our rebuttal has addressed all your concerns. If there are still aspects that need further clarification, please feel free to continue the discussion with us!
>
> > ***Weakness #1: The invertibility is not perfect, and how to address the artifacts.***
>
> - We acknowledge that achieving perfect invertibility through DDIM Inversion can be challenging. Therefore, at this stage, we only require the revealed image to match the secret image in subjective visual quality. By ensuring **acceptable subjective fidelity**, we can fully exploit the advantages of CRoSS in terms of security and robustness.
> - Regarding the concern you raised about "low-level artifacts", we suspect you are referring to the situation where some of the revealed images might appear to have slightly lower quality. This phenomenon can be attributed to the generation capability of the diffusion model, which directly impacts the quality of the revealed image. However, it's important to note that our method itself is **independent of the specific choice of the diffusion model**. Opting for a more powerful diffusion model (such as the latest SDXL [1]) can further enhance the quality of the revealed image, thereby potentially alleviating the concern you mentioned.
>
> > ***Weakness #2 & Question #1: The similarity between the secret and container images' backgrounds***
>
> - In our experimental setting, there is indeed a similarity in background between the secret image and the container image. This similarity is primarily influenced by the editing capability of Stable Diffusion (with guidance scale = 1) that we employ. Therefore, your observation is correct: a better editor could enhance our proposed CRoSS framework in a plug-and-play manner, resulting in greater diversity between the secret image and the container image.
>
> > ***Weakness #3: The generation capability and stability on human face images***
>
> - Based on our knowledge, Stable Diffusion does possess the ability to generate high-quality human faces. There are numerous galleries within the community that serve as references, such as the following blog [2].
> - Furthermore, within our experimental setting, we do not solely rely on Stable Diffusion to generate results from scratch. Instead, we utilize DDIM Inversion for image editing, which is less challenging and exhibits higher stability.
>
> > ***References***
>
> [1] https://stablediffusionxl.com/
>
> [2] https://www.reddit.com/r/StableDiffusion/comments/13bjs6x/some_unedited_faces_made_with_base_sd_15/

---

> ### Author Response · Authors · 2023-08-18
> **Looking forward to discussions with the Reviewer Nci3**
>
> Dear Reviewer Nci3,
>
> We appreciate the time you dedicated to reviewing our work and your recognition of our work. Regarding the concerns you raised, we have provided explanations in our responses.
>
> We would like to ensure that your concerns have been adequately addressed. If there are any aspects of our work that remain unclear to you, please don't hesitate to let us know.
>
> Thank you for your dedication!
>
> Best regards,
>
> Authors

---

### Official Review · Reviewer_mtD2 · 2023-08-02

**Soundness:** 3 good
**Presentation:** 3 good
**Contribution:** 3 good
**Rating:** 6
**Confidence:** 3

**Summary:**

In this work, the authors introduce an image steganography framework (CRoSS) that leverages the properties of diffusion models to enhance the security, controllability, and robustness of the steganography process. The authors show how the diffusion model can integrate with image steganography to achieve these goals without additional training. The authors claim this to be the first work to introduce diffusion models to the field of image steganography. The effectiveness of the proposed CRoSS framework was validated with different experiments where the authors demonstrated the advantages over existing methods.


**Strengths:**

- This paper proposes to apply diffusion models to the field of image steganography, creatively combining existing ideas to overcome the limitations of traditional methods.

- The paper is clear, and the authors clearly define their goals, explain their methodology, and discuss their results.

- Overall, the paper was easily understandable and easy to follow. Its structure is well-organized, and the proposed method is presented clearly and makes sense.

- To validate the effectiveness of the method, the authors conducted different experiments showing the advantages of the proposed approach in terms of security, controllability, and robustness.

- In general, the paper is very well written.

**Weaknesses:**

- Although the authors considered validating their method against distortion attacks, Gaussian noise distortion is not the appropriate approach to evaluate its robustness. As authors may know,  Gaussian noise is already used in the diffusion process due to its mathematical properties. The Gaussian distribution is symmetric and has the property that the sum of multiple Gaussian random variables is also Gaussian. This makes it mathematically convenient, especially in diffusion models where noise is added iteratively. Therefore, adding more Gaussian noise to the container image will not affect the reveal process too much. I would like to see if the method is still robust to another type of noise but Gaussian.
- In my opinion, the robustness validation made in the paper is not enough to conclude that the method is robust. For example, did the authors consider the scenario where an attacker performs not only global distortions (like JPEG compression, adding noise, etc.) but local distortions, e.g., blurring a patch on the image?

**Questions:**

- Will the proposed method recover the secret image well when an attacker applies another type of noise? Can this negatively impact the performance of the method?
- Is the method robust to local distortions?

**Limitations:**

The authors acknowledged the limitations of their work, including the gap in pixel-wise objective fidelity metrics, the trade-off that sacrifices the editing capability to ensure the zero-shot invertibility of image translation, and the limitation of hiding only one secret image within a single container image. However, they could provide more insights into potential solutions or future research directions to address these limitations. They did not explicitly discuss potential negative societal impacts.

---

> ### Author Rebuttal · Authors · 2023-08-06
>
> Thank you for your constructive comments! We hope that our response addresses all of your concerns. All discussions and supplementary experiments will be included in our revised version. If there are any remaining questions that have not been resolved, please feel free to continue the discussion with us!
>
> $\textbf{\textcolor{red}{The supplementary rebuttal PDF file can be found at the bottom of the overall response.}}$
>
> > ***Weakness #1 & Question #1: Robustness experiments involving a broader range of noise types***
>
> - The robustness experiments involving additional types of noise are provided in the supplementary rebuttal PDF file (**Table 1 and Figure 3**). We have included the results for **Poisson noise** and **salt-and-pepper noise**. The experimental results show that CRoSS maintains robustness against other types of noise and has an advantage compared to other methods.
> - Regarding the implementation details of the two types of noise, for Poisson noise, we referred to the implementation in [1] and adjusted the noise level using the parameter $\alpha$. For salt-and-pepper noise, we adjusted the noise level using the parameter probability $p$. The examples of degraded images are presented in **Figure 2** of the Supplementary Rebuttal PDF file.
>
> > ***Weakness #2 & Question #2: Robustness experiments involving local distortions***
>
> - The robustness experiments concerning local distortions are presented in the supplementary rebuttal PDF file (**Figure 1**). We have included the results for the scenario of **blurring a patch**. The experimental results indicate that CRoSS maintains significant robustness against local distortions and exhibits clear advantages compared to other methods.
> - Regarding the implementation details of "blurring-a-patch," we extracted a $256\times256$ patch from the center of the image and applied Gaussian blur exclusively to this patch. The blur kernel size was set to $5$, and the sigma was set to $2$. The examples of degraded images are presented in **Figure 2** of the Supplementary Rebuttal PDF file.
>
> > ***Future research directions for limitations***
>
> - (1) Regarding the limitation of **the gap in pixel-wise objective fidelity metrics**, which is primarily attributed to the training-free nature of DDIM Inversion, a potential solution could involve training diffusion models specifically for image steganography tasks to improve objective fidelity.
> - (2) Regarding the limitation of **sacrificing editing capability to ensure invertibility**, which is mainly attributed to the unsatisfactory generation capabilities of Stable Diffusion (with guidance scale = 1), a potential strategy involves keeping up with the latest advancements in the diffusion model field and replacing the base model in CRoSS in a plug-and-play way to alleviate this issue.
> - (3) Regarding the limitation of **hiding only one secret image within a single container image**, this is primarily due to the current design of diffusion models being Single Input Single Output (SISO). A potential solution could involve designing Multiple Input Multiple Output (MIMO) diffusion models specifically for high-capacity image steganography.
>
> > ***Potential negative societal impacts***
>
> - The related technology could potentially be utilized for improper distribution of personal privacy or for inappropriate distribution of images that might lead to offense.
>
> > ***References***
>
> [1] Practical Blind Denoising via Swin-Conv-UNet and Data Synthesis. ArXiv 2022.

---

> > ### Comment · Reviewer_mtD2 · 2023-08-11
> >
> > I thank the authors for their detailed response. The provided rebuttal addressed my questions. I will update and maintain my score toward acceptance.

---

> > > ### Author Response · Authors · 2023-08-12
> > > **Thank Reviewer mtD2 for recognizing our work**
> > >
> > > Dear Reviewer mtD2,
> > >
> > > Thank you for engaging in our discussion and recognizing that our responses have effectively addressed all the concerns you raised. We greatly appreciate your constructive comments, which have contributed to the improvement and solidity of our work!
> > >
> > > Best regards,
> > >
> > > Authors

---

### Author Rebuttal · Authors · 2023-08-06

We sincerely appreciate all the constructive comments from the reviewers! Below is our brief overall response.

> ***Firstly, we are delighted to observe that the reviewers have acknowledged various aspects of our work:***

- Reviewer mtD2 and Reviewer Nci3 hold a $\textbf{\textcolor{red}{positive}}$ side for our work in terms of $\textbf{\textcolor{red}{soundness, presentation, contribution, novelty and effectiveness.}}$
- $\textbf{\textcolor{red}{All}}$ reviewers, including Reviewer mtD2, Reviewer Nci3, Reviewer g14p, and Reviewer 1iwK, consider our attempt to introduce diffusion models into the field of image steganography as a $\textbf{\textcolor{red}{strength}}$ of our work.

> ***Secondly, we would like to emphasize the value and contribution of our work:***

- Based on diffusion models, CRoSS demonstrates unique advantages in terms of $\textbf{\textcolor{red}{security and robustness}}$, while maintaining acceptable subjective fidelity. This highlights the $\textbf{\textcolor{red}{greater practicality}}$ of CRoSS compared to previous non-diffusion model methods.
- CRoSS introduces the $\textbf{\textcolor{red}{first}}$ image steganography framework based on diffusion models, and this framework exhibits strong extensibility, serving as $\textbf{\textcolor{red}{inspiration}}$ for the future development of the field of image steganography.
- We have observed that Reviewer g14p claims a lack of novelty and insights in our work. However, the advantages and contributions listed above clearly demonstrate the novelty of CRoSS.

> ***Thirdly, we would like to further clarify the concerns regarding objective fidelity and practicality.***

- We have observed Reviewer 1iwK's concerns regarding the objective fidelity and practicality of CRoSS. However, evaluating an image steganography algorithm requires $\textbf{\textcolor{red}{a comprehensive consideration of fidelity, security, and robustness}}$, which many previous methods have overlooked, leading to poor practicality. In contrast, CRoSS exhibits excellent security and robustness, along with acceptable subjective fidelity, which grants it significant practical advantages.

We kindly request the reviewers to thoroughly consider the value and contribution of our work. The detailed rebuttals for each reviewer can be found below.

Additionally, we have attached a $\textbf{\textcolor{red}{PDF file}}$ containing some figures and tables for the reviewers' reference.

---

### Decision · Program_Chairs · 2023-09-21

**Decision:**

Accept (poster)

**Comment:**

In this paper, the authors introduce an image steganography framework that uses a diffusion model to enhance controllability, and robustness of the steganography method. It is a significant benefit that this can be achieved using a foundation diffusion model and does not require additional training.

One reviewer maintained a concern that the comparison between RIIS and CROSS is not fair, because the baseline method has not been trained with Stable Diffusion. There was extensive discussion but the reviewer remained concerned.  This is understood but leveraging powerful pre-trained models is beneficial if it improves performance, so despite the fairness issue this is progress in my opinion.

There were additional concerns on the image quality for some of the experiments but overall the paper seems to have good ideas and reasonable experiments so I recommend acceptance.